# ATR/Mec1 prevents lethal meiotic recombination initiation on partially replicated chromosomes in budding yeast

Hannah G Blitzblau[1], Andreas Hochwagen[1,2]*

[1]Whitehead Institute for Biomedical Research, Cambridge, United States;
[2]Department of Biology, New York University, New York, United States

**Abstract** During gamete formation, crossover recombination must occur on replicated DNA to ensure proper chromosome segregation in the first meiotic division. We identified a Mec1/ATR- and Dbf4-dependent replication checkpoint in budding yeast that prevents the earliest stage of recombination, the programmed induction of DNA double-strand breaks (DSBs), when pre-meiotic DNA replication was delayed. The checkpoint acts through three complementary mechanisms: inhibition of Mer2 phosphorylation by Dbf4-dependent Cdc7 kinase, preclusion of chromosomal loading of Rec114 and Mre11, and lowered abundance of the Spo11 nuclease. Without this checkpoint, cells formed DSBs on partially replicated chromosomes. Importantly, such DSBs frequently failed to be repaired and impeded further DNA synthesis, leading to a rapid loss in cell viability. We conclude that a checkpoint-dependent constraint of DSB formation to duplicated DNA is critical not only for meiotic chromosome assortment, but also to protect genome integrity during gametogenesis.

*For correspondence: andi@nyu.edu

**Competing interests:** The authors declare that no competing interests exist.

**Reviewing editor**: Johannes Walter, Harvard Medical School, United States

## Introduction

During meiosis, a single round of DNA replication is followed by two nuclear divisions to produce haploid gametes from diploid progenitor cells. In most organisms, the faithful segregation of homologous chromosome pairs in meiosis I relies on physical connections between homologs produced by meiotic recombination. In budding yeast, DNA exchanges between homologs are the result of the repair of ~160 DNA double-strand breaks (DSBs), which occur immediately following pre-meiotic S phase (meiS) and are distributed across all 16 chromosome pairs. DSB formation requires the concerted activity of 10 proteins, including the catalytic subunit Spo11 and its partner Ski8, the Mre11/Rad50/Xrs2 (MRX) complex, and the meiosis-specific proteins Mer2, Rec114, Mei4, Rec102 and Rec104 (reviewed in (*Keeney and Neale, 2006*)). Additionally, DSB formation requires the phosphorylation of Mer2 by two cell cycle kinases, cyclin dependent-kinase (CDK) and the Dbf4-dependent Cdc7 kinase (DDK) (*Henderson et al., 2006*; *Sasanuma et al., 2008*; *Wan et al., 2008*). The complexity of this process reflects the fact that meiotic genome fragmentation needs to be carefully controlled to limit genome instability.

Under normal circumstances, meiotic DSBs are introduced after DNA replication. This temporal separation is necessary because the crossover-mediated linkages between homologs require sister-chromatid cohesion distal to the crossover site. Thus, only crossovers formed after DNA replication serve to hold homologous chromosomes together in metaphase I. Moreover, replication forks are unable to cross a DSB (*Doksani et al., 2009*), so the presence of >100 DSBs in the genome would severely interfere with the completion of DNA replication.

The mechanisms that coordinate pre-meiotic DNA replication and DSB formation are not well understood. DNA replication and DSB formation are coordinated at the local level, because delayed replication of a single chromosome arm delays DSB formation on that arm (*Borde et al., 2000*).

**eLife digest** Most cells in an organism contain two sets of chromosomes, one inherited from the mother and the other from the father. However, sexual reproduction relies on the production of gametes—eggs and sperm—which contain only one set of chromosomes. These are produced through a specialized form of cell division called meiosis.

Meiosis begins with a cell replicating its entire genome. Maternal and paternal versions of each chromosome then pair up and swap sections of their DNA through a process known as homologous recombination. This gives rise to chromosomes with new combinations of maternal and paternal genes. Finally, the cell undergoes two successive rounds of division—the first to produce a cell with two nuclei containing two sets of chromosomes each, and the second to produce four gametes, each containing a single set of chromosomes.

Homologous recombination requires the formation of double-strand breaks in the DNA, but it is essential that these do not form before DNA replication is complete. Now, Blitzblau and Hochwagen have used yeast, which is easy to maintain in the lab and to manipulate genetically, to reveal the molecular components of a checkpoint that controls this process.

Blitzblau and Hochwagen first used an inhibitor called hydroxyurea to block DNA replication in yeast cells, and confirmed that this treatment also suppressed the formation of double-strand breaks. By selectively inhibiting the activity of individual proteins, it was shown that break formation was controlled by a checkpoint that relies on two conserved proteins, the checkpoint kinase Mec1 (homologous to the human tumour suppressor ATR) and the cell-division kinase DDK. Moreover, when double-strand breaks were allowed to form on partially replicated chromosomes, DNA replication stalled and meiosis could not proceed normally, with lethal results for the yeast.

These results explain how DNA replication and recombination are coordinated during meiosis in yeast. Moreover, because the genes that control meiosis are highly conserved from yeast to humans, they have implications for research into human fertility.

However, DSB formation does not require DNA replication; pre-meiotic replication initiation mutants introduce full levels of DSBs on chromosomes that are not replicated in both budding and fission yeasts (*Murakami and Nurse, 2001*; *Hochwagen et al., 2005*; *Blitzblau et al., 2012*). In addition, the initiation of meiotic recombination is prevented globally when DNA replication is delayed by nucleotide depletion (*Schild and Byers, 1978*; *Tonami et al., 2005*; *Ogino and Masai, 2006*). In fission yeast, a replication checkpoint blocks DSB formation in this situation (*Tonami et al., 2005*; *Ogino and Masai, 2006*). Although a related checkpoint was found to delay the meiotic divisions upon nucleotide depletion in budding yeast (*Stuart and Wittenberg, 1998*), a subsequent study came to the conclusion that it did not regulate DSB formation (*Borde et al., 2000*). Thus, it remains unclear how budding yeast prevent DSB formation on unreplicated DNA.

The replication checkpoint couples DNA replication and cell cycle progression by sensing and coordinating the response to delayed replication forks (reviewed in (*Labib and De Piccoli, 2011*). In vegetatively growing yeast cells, stalled replication forks activate a conserved kinase cascade including Mec1/ATR and Rad53/CHK2. Mec1 and Rad53 inhibit cell cycle progression by preventing chromosome segregation and mitotic entry, respectively (*Clarke et al., 2001*; *Clarke et al., 2003*). Additionally, their activation stabilizes replication forks, preventing catastrophic fork collapse. Finally, activated Rad53 also phosphorylates and directly interacts with the Dbf4 subunit of DDK (*Weinreich and Stillman, 1999*; *Duncker et al., 2002*; *Chen et al., 2013*), which delays further initiation of DNA replication.

The replication checkpoint has not been characterized in budding yeast meiosis. It presumably functions, because cells can respond to and recover from replication inhibition (*Schild and Byers, 1978*; *Blitzblau et al., 2012*) and meiotic functions of both Mec1 and Rad53 have been described (reviewed in (*MacQueen and Hochwagen, 2011*)). The fact that the replication checkpoint inhibits DDK activity in mitotic cells suggests that the checkpoint could be easily adapted to prevent meiotic DSBs by preventing the DDK-dependent phosphorylation of Mer2. To characterize the replication checkpoint during meiotic cell division, we investigated the effects of inhibiting pre-meiotic DNA replication. We found that the replication checkpoint is active in budding yeast meiosis and inhibits

DSB formation. The checkpoint uses parallel mechanisms to regulate the abundance, DNA loading and DDK-dependent phosphorylation of DSB factors. Cells that formed DSBs on partially replicated chromosomes were unable to complete either DSB repair or genome duplication, revealing that the separation of DNA replication and meiotic DSB formation is critical for maintaining genome integrity and viability.

## Results

### Inhibited DNA replication delays DSB formation

To determine whether DSB formation is coordinated with bulk DNA replication during meiS in budding yeast, we exposed cells to increasing doses of the replication inhibitor hydroxyurea (HU) and measured the kinetics of DNA replication and DSB formation (using a *rad50S* mutation to prevent DSB repair). Because we previously observed significantly delayed meiotic entry when cells were treated with high concentrations of HU (*Blitzblau et al., 2012*), all of our analyses were carried out with 20 mM or lower amounts of HU. FACS analysis of total DNA content revealed that DNA replication occurred between 1–3 hr for wild-type cells in the absence of HU, was significantly delayed in 5 mM HU, and arrested in early S phase in 20 mM HU (*Figure 1A*). DSB formation was comparably affected when measured by Southern blot analysis of a prominent DSB hotspot on chromosome 3 (*Figure 1B*). Quantification of FACS profiles and Southern analysis revealed that DSBs appeared just after bulk DNA replication was completed (4C DNA content appeared) in 0 or 5 mM HU samples, and were fully suppressed when replication was blocked by 20 mM HU (*Figure 1C*). Consistent with the idea that slowing DNA replication activates the replication checkpoint, we detected HU-dependent Rad53 autophosphorylation by Western blotting (*Figure 1D*), which has been shown to be a direct effect of checkpoint activation in pre-mitotic cells (*Pellicioli et al., 1999*). In addition, we found that Mec1 and Rad53 were essential to maintain viability in HU-treated pre-meiotic cells (*Figure 1—figure supplement 1A*), indicating that activation of the pre-meiotic replication checkpoint is critical to maintain replication forks in the presence of replication inhibition, as in pre-mitotic cells.

We tested whether the DSB delay in HU-treated cells was due to inhibited DNA replication, as the mitotic replication checkpoint is characteristically activated by ssDNA at the replication fork. DNA replication is strongly decreased in *cdc6-mn* cells (*Hochwagen et al., 2005*; *Blitzblau et al., 2012*). This decrease was due to impaired replicative helicase loading (*Figure 1—figure supplement 1B*), and little DNA replication was observed in 0, 5 or 20 mM HU (*Figure 1E*). The *cdc6-mn* cells formed DSBs with wild-type kinetics in all concentrations of HU despite the absence of any completed DNA replication (*Figure 1F,G*), consistent with the idea that depleting the number of active replication forks abrogates the replication checkpoint signal. Importantly, we were unable to detect phosphorylation of Rad53 at 2–3 hr when DSBs formed (*Figure 1H*), indicating that the replication checkpoint is not efficiently activated in these cells. When *cdc6-mn* cells were exposed to concentrations of HU greater than 20 mM, DSB formation was either reduced or abolished without activation of Rad53 (data not shown), consistent with our previous report that high levels of HU can inhibit meiotic cell cycle entry (*Blitzblau et al., 2012*). This could explain why the checkpoint was previously not observed (*Borde et al., 2000*). Together, these data confirm that the canonical Mec1- and Rad53-dependent replication checkpoint responds to delayed DNA replication in budding yeast meiosis, and that DSB formation is delayed while DNA replication is ongoing.

### Mec1 signals to inhibit Mer2 phosphorylation by DDK

Given that Rad53 inhibits DDK in mitotically dividing cells and that DDK activates the meiotic DSB factor Mer2, we explored whether inhibition of pre-meiotic DNA replication delayed phosphorylation of Mer2. As shown in *Figure 2A*, when pre-meiotic cells were treated with HU, Dbf4 accumulated mainly in a hyperphosphorylated state. The amount of hyperphosphorylated Dbf4 was reduced in both *mec1Δ* and *rad53Δ* cells treated with HU (top panel), indicating that the massive accumulation of this form of the protein is checkpoint-dependent in meiotic cells as it is in mitotic cells (*Weinreich and Stillman, 1999*). For this analysis we used a polyclonal antibody to Dbf4, having noted that the C-terminally myc-tagged protein was unstable and present at much lower levels and with degradation products, compared to the untagged protein (data not shown). Consistent with the idea that the hyperphosphorylated form of Dbf4 is inactive in the cell, Mer2 accumulated in a hypophosphorylated form in HU-treated cells (*Figure 2A*, bottom panel) only when Mec1 and Rad53 were present. Because

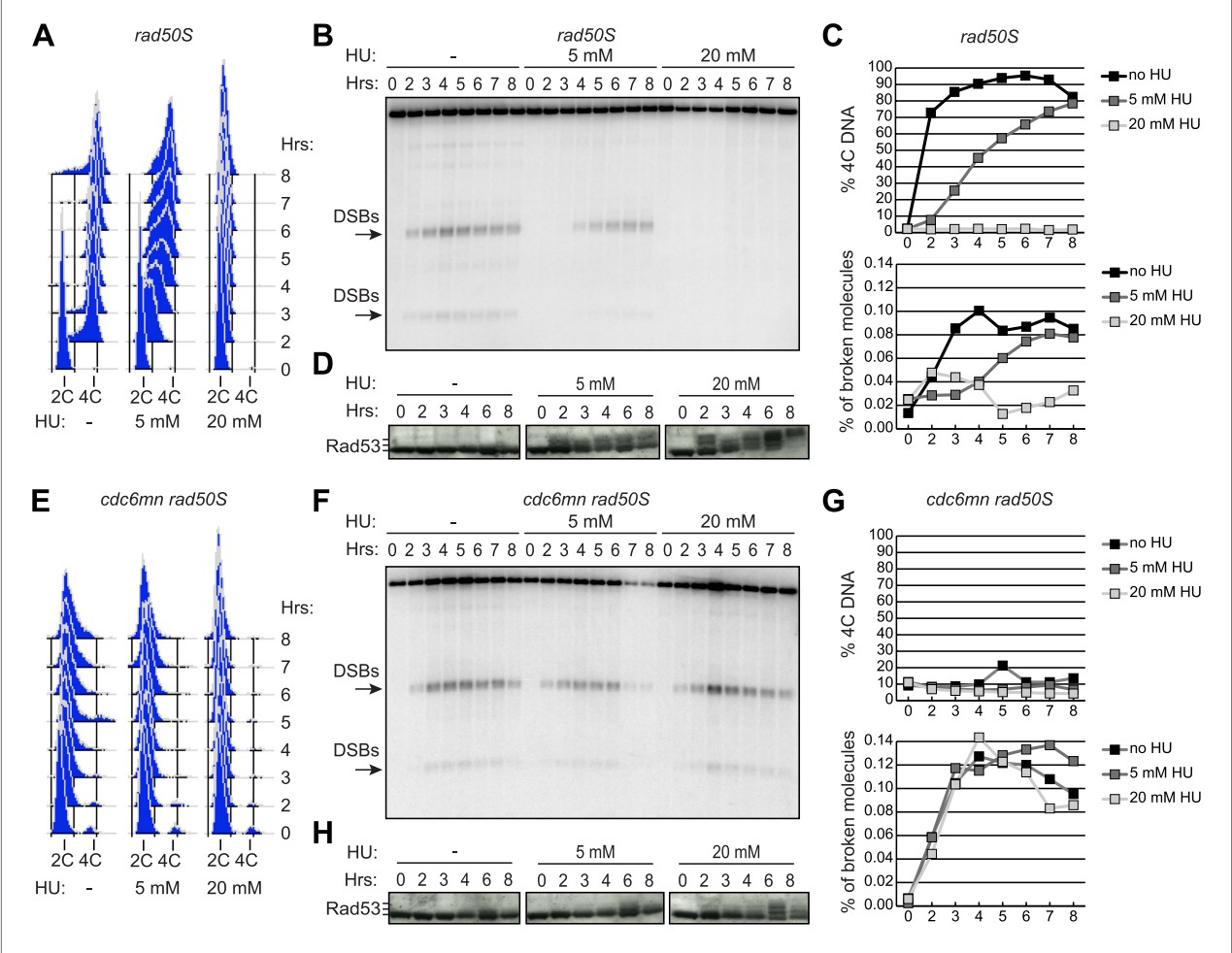

**Figure 1**. Ongoing DNA replication delays meiotic DSB formation. *rad50S* (H156, **A**–**D**) or *cdc6-mn rad50S* (H155, **E**–**H**) cells were induced to enter meiosis in 0, 5 or 20 mM HU and analyzed at the indicated time points. (**A** and **E**) FACS analysis of total DNA content. (**B** and **F**) Southern blot analysis of DSB formation at the *yCR048W* DSB hotspot. Arrows indicate the major DSB bands quantified in (**C** and **G**). (**C** and **G**) Quantification of 4C DNA content from FACS is shown in the upper panel. The measurement of DSBs from Southern blot is plotted in the lower panel. (**D** and **H**) Western blot analysis of Rad53 protein mobility is shown as a measurement of phosphorylation and activation. Slower migrating bands correspond to phosphorylated Rad53. See also *Figure 1—figure supplement 1*.

The following figure supplements are available for figure 1:

**Figure supplement 1**. Replication and checkpoint requirements during meiS.

Mer2 is sequentially phosphorylated by both CDK and DDK prior to meiotic DSB formation (*Sasanuma et al., 2008*; *Wan et al., 2008*), we confirmed that Mer2 was phosphorylated by CDK in HU-treated cells (*Figure 2—figure supplement 1A,B*), indicating that HU treatment specifically inhibited DDK-dependent phosphorylation of Mer2. These results indicate that the replication checkpoint kinases Mec1 and Rad53 restrain DSB formation by limiting the activity of DDK, a role well established in the mitotic replication checkpoint.

To test whether the activation of DDK is sufficient to allow for DSB formation in pre-meiotic cells, we sought an unregulated version of Dbf4 that could allow for DSB formation in HU-treated cells. We tested several previously characterized alleles of *DBF4* in our Southern blot assay for DSBs (*Figure 2B,C*, *Figure 2—figure supplement 1*). Cells expressing *dbf4-NLSΔN221*, an N-terminal truncation protein containing an SV40 nuclear localization signal (NLS) to support viability, formed high levels of DSBs in the presence of HU (*Figure 2C*) without a substantial increase in DNA replication (*Figure 2D*). The replication checkpoint bypass was dominant as DSBs also occurred in cells heterozygous for this *dbf4*

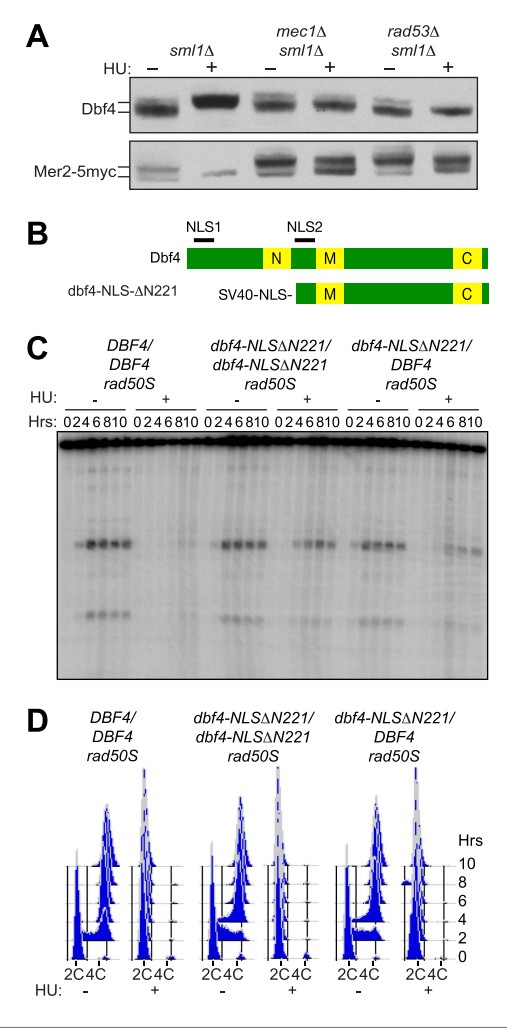

**Figure 2**. The pre-meiotic replication checkpoint inhibits DDK kinase activity. (**A**) Western blot analysis of Dbf4 (top panel) and Mer2-5myc (bottom panel) in *sml1Δ* (H5157), *mec1Δ sml1Δ* (H5220) and *rad53Δ sml1Δ* (H5127) cells. The *sml1Δ* mutation was used to maintain viability of *mec1Δ* and *rad53Δ* mutants. For Mer2-5myc blotting, only 20% (wild-type) or 50% (*mec1Δ* and *rad53Δ*) of total protein was loaded for HU-treated samples as high accumulation of the Mer2 protein obscured the analysis of mobility shifts. (**B**) Schematic of wild-type and mutant Dbf4 proteins analyzed in this study. N, M, and C refer to the N-terminal, middle and C-terminal conserved domains. (**C**) Southern blot analysis of DSB formation at the *yCR048W* DSB hotspot in *rad50S* cells for *DBF4/DBF4* (H6097), *dbf4-NLS-ΔN221/dbf4-NLS-ΔN221* (H6146) and *dbf4-NLS-ΔN221/DBF4* (H7335) cells. (**D**) FACS analysis of total DNA content of the strains in (**C**). See also *Figure 2—figure supplement 1*.

The following figure supplements are available for figure 2:

**Figure supplement 1**. Regulation of DDK prevents DSBs in HU-treated cells.

allele (*Figure 2C*). This result indicates that deregulating DDK activity is sufficient to allow the initiation of meiotic recombination in the presence of ongoing DNA replication. The dbf4-NLSΔN221 protein lacks the conserved N domain that has been shown to interact with Rad53 and other proteins (*Gabrielse et al., 2006*; *Chen et al., 2013*). However, cells harboring a smaller truncation of *DBF4* that maintains the native NLS but removes the Rad53 interaction domain, *dbf4-Δ71-221*, bypassed the replication checkpoint poorly (*Figure 2—figure supplement 1C*), indicating that disrupting the Rad53 interaction alone is not sufficient to deregulate Dbf4 activity. Similarly, addition of the SV40 NLS to the wild-type Dbf4 protein did not allow for DSBs in HU-treated cells (*Figure 2—figure supplement 1D*), indicating the strong NLS was not solely responsible for the replication checkpoint bypass of the *dbf4-NLSΔN221* allele. Thus, multiple functions of the Dbf4 N-terminus are likely required for the regulation of meiotic DSBs in response replication stress. We conclude that DDK regulation is critical in the pre-meiotic replication checkpoint.

Given that Dbf4 is regulated by Rad53-dependent phosphorylation, we explored the role of such phosphorylation in preventing meiotic DSBs in response to HU-treatment. A *dbf4* allele lacking 25 potential phosphorylation sites, *dbf4-m25*, allows the initiation of DNA replication from late origins in HU-treated pre-mitotic cells (*Lopez-Mosqueda et al., 2010*). However, this mutation did not permit DSB formation in pre-meiotic cells treated with HU (*Figure 2—figure supplement 1E*). This result indicates that simply preventing the Rad53-dependent phosphorylation of Dbf4 is insufficient to accumulate enough DDK activity to form meiotic DSBs in HU-treated cells. Furthermore, the *dbf4-NLSΔN221* allele produced a truncated protein that shifted mobility consistent with phosphorylation in the presence of HU (*Figure 2—figure supplement 1F*), despite the fact that 21 of the 25 mutations in the *dbf4-m25* allele are within the deleted region. These results suggest either the *dbf4-NLSΔN221* mutant protein is unable to respond to Rad53-dependent phosphorylation, or that separate phosphorylation sites are required to prevent meiotic DSBs. We ruled out a function for the sole Mec1 consensus site on Dbf4, because mutation of threonine 163 to alanine did not allow DSB formation in HU (*Figure 2—figure supplement 1G*). Together, these data reveal that deregulating DDK is sufficient to allow DSBs in HU-treated cells. However, simply preventing the phosphorylation of Dbf4 or

its interaction with Rad53 does not produce enough DDK activity to allow for DSB formation in HU-treated cells.

## Mec1 inhibits meiotic DSBs in a Rad53-independent manner

Given that Dbf4 activity is critical for DSB formation, we tested whether removing the checkpoint kinases Mec1 and Rad53 that control Dbf4 activity was sufficient to allow DSB formation in HU-treated cells. We found that disruption of Mec1, either by deletion or using the *mec1-1* allele (**Figure 3A**, **Figure 3—figure supplement 1** and **Figure 4—figure supplement 1**), or removal of the Mec1-interacting protein Ddc2 (**Figure 3—figure supplement 1A**) was sufficient to allow DSB formation in HU-treated cells. DSB formation in *mec1Δ* cells occurred along entire chromosomes (**Figure 3—figure supplement 1B**) without significantly increased DNA replication (**Figure 3B**, **Figure 3—figure supplement 1C**), indicating the replication checkpoint was bypassed. The levels of DSB formation in *mec1Δ* cells treated with HU were lower than those observed in the absence of HU (**Figure 3C**), which we believe is due to relocalization of DSB factors in checkpoint mutants (discussed in more detail later). In contrast, *rad53Δ* cells failed to form DSBs in the presence of HU (**Figure 3**, **Figure 3—figure supplement 1C**). These results were confirmed in *dmc1Δ* repair-deficient strains, which arrest in meiotic

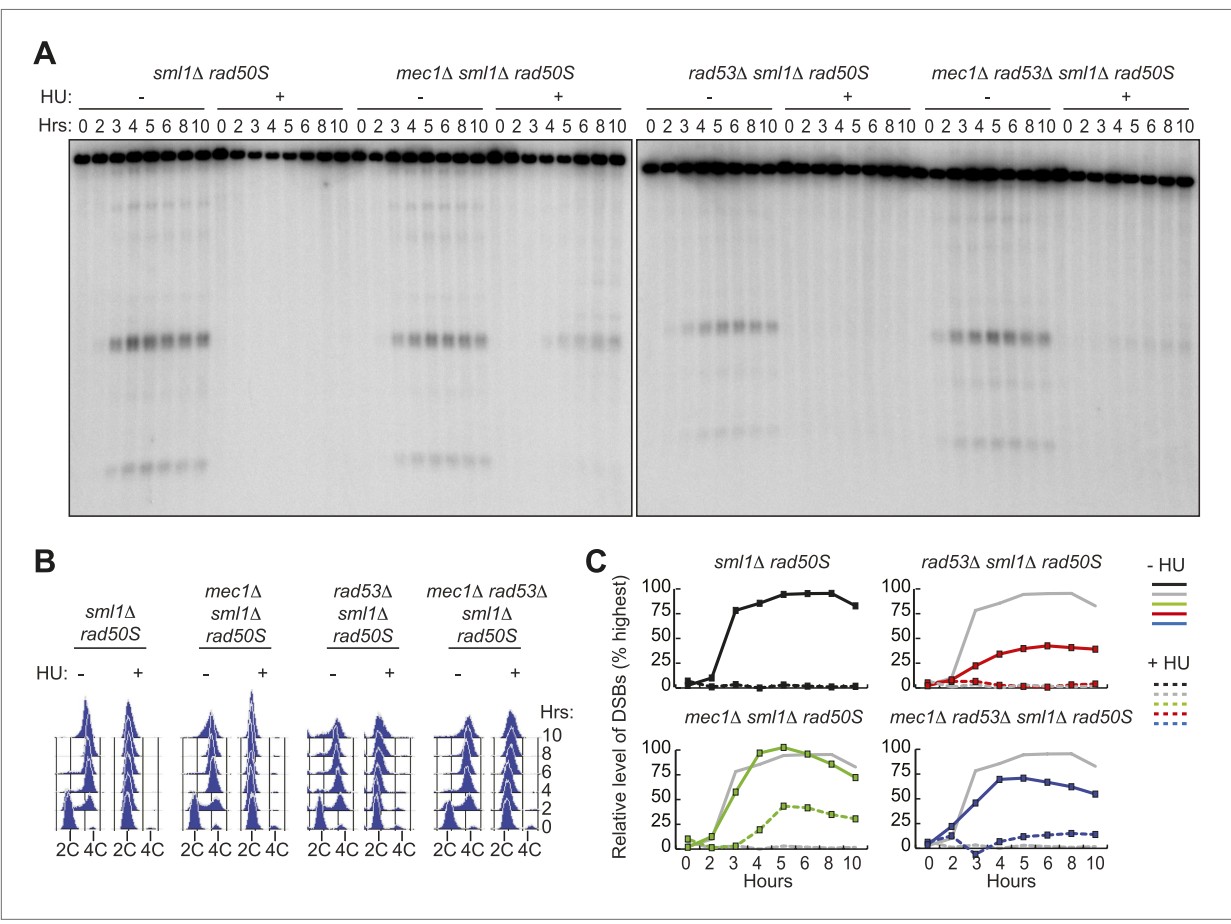

**Figure 3**. Removal of *MEC1*, but not *RAD53*, allows DSB formation in HU-treated cells. (**A**) Southern blot analysis of DSB formation at the *yCR048W* DSB hotspot in *sml1Δ rad50S* (H4898), *mec1Δ sml1Δ rad50S* (H4935) and *rad53Δ sml1Δ rad50S* (H4969) and *mec1Δ rad53Δ sml1Δ rad50S* cells (H4932) in the absence or presence of HU. (**B**) FACS analysis of total DNA content of the strains in (**A**). (**C**) Quantification of the relative DSB levels from the Southern blot in (**A**) in the absence (solid lines) or presence (dashed lines) of HU. DSBs levels were measured as in **Figure 1** and normalized to the maximum measurement in the *sml1Δ rad50S* (H4898) 'wild-type' strain (shown as grey lines for comparison). See also **Figure 3—figure supplement 1**.

The following figure supplements are available for figure 3:

**Figure supplement 1**. Removing Mec1, but not Rad53, allows DSBs in HU-treated cells.

prophase with resected DNA ends (*Figure 3—figure supplement 1D*), indicating they were not specific to the *rad50S* allele used in our initial assays. We eliminated the possibility that the pre-meiotic replication checkpoint utilizes alternate signaling kinases by simultaneously removing Mek1, Chk1 and Rad53, which did not allow DSB formation in HU-treated cells (*Figure 3—figure supplement 1E*). Similarly, neither removal of Rad53 kinase activity nor the deletion of the Rad53 activating proteins Rad9 and Mrc1 allowed DSBs to form in HU-treated cells (*Figure 3—figure supplement 1F,G*). Indeed, removal of Mec1, but not Rad53, Rad53 kinase activity or Rad53-dependent phosphorylation sites on Dbf4, allowed DSB formation across entire chromosomes in both *rad50S* and *dmc1Δ* repair-deficient strains (*Figure 3—figure supplement 1B,D*). We noted that the lack of checkpoint bypass in the absence of Rad53 is in contrast to the bypass observed in Dbf4 truncation mutants (*Figure 2*). These findings argue against a strictly linear pathway, in which Mec1 acts through Rad53 to inhibit DDK, but rather suggest that there are Rad53-independent functions of the replication checkpoint in meiosis.

## A Tel1-dependent pathway increases DSB levels in *rad50S* cells

In the course of carrying out these experiments, we noticed that *rad53Δ* and *rad53-kd* cells exhibited lower DSB levels in untreated cells, specifically in the *rad50S* background (*Figures 3 and 4A*, *Figure 3—figure supplement 1C,D*), suggesting that Rad53 may play a second role in promoting meiotic DSBs in this context. A DSB-promoting role for Rad53 is also supported by our observation that deletion of *RAD53* reduced the DSB levels of *mec1Δ* cells (*Figure 3*). In this case, DSBs were formed in HU-treated *mec1Δ rad53Δ* cells, but the break levels were substantially lower than in *mec1Δ* cells alone (*Figures 3A,C and 4A*), indicating Mec1 and Rad53 regulate DSBs using separate pathways. The *rad50S* mutation leads to the formation of blunt-ended DSBs that are detected by a Tel1 and Rad53-dependent checkpoint (*Xu et al., 1997*; *Usui et al., 2001*). We found that removal of Tel1 also lowered DSB levels in the absence of HU at the *yCR048W* hotspot (*Figure 4A*, *Figure 4—figure supplement 1*) and on whole chromosomes analyzed by pulsed-field gel analysis (*Figure 4B*), consistent with a recent report (*Argunhan et al., 2013*). Similar to the *rad53Δ* mutation, deletion of *TEL1* also reduced DSBs in a *mec1–1* background, while still allowing for replication checkpoint bypass in the presence of HU (*Figure 4—figure supplement 1*). Intriguingly, DSBs levels were also lowered in *dbf4-NLSΔN221* cells (*Figure 4A*), but not *dbf4-m25* mutants (*Figure 2—figure supplement 1D*), suggesting that Rad53 might promote DDK activity in a phosphorylation-independent manner. Thus, Tel1 and Rad53 are required to achieve maximal DSB levels in *rad50S* cells, whereas Mec1 inhibits DSBs in response to delayed replication using both Rad53-dependent and -independent mechanisms. These data extend recent findings that multiple checkpoint pathways modulate DSB formation in budding yeast (*Zhang et al., 2011*; *Argunhan et al., 2013*; *Carballo et al., 2013*; *Gray et al., 2013*).

## Mec1 inhibits the DNA loading of DSB factors

To address the mechanism of action of the pre-meiotic replication checkpoint, we analyzed DSB factor abundance and modifications in the absence or presence of HU, as DSB factor accumulation is limited by the checkpoint in *Schizosaccharomyces pombe* (*Ogino and Masai, 2006*). To ensure that the modifications we observed were the result of the replication checkpoint and not of DSBs themselves (which also activate Mec1), the analysis was carried out in *spo11-Y135F ndt80Δ* mutants that arrest in meiotic prophase without DSBs. The protein levels of most DSB factors were unchanged in the absence or presence of HU (*Figure 5—figure supplement 1A*), although we found that Mer2 accumulated to high levels in HU-treated cells (*Figure 2A* and legend, *Figure 2—figure supplement 1A* and *Figure 5—figure supplement 1A*).

The exceptional protein in this analysis was Spo11, whose levels were approximately 10-fold lower in HU-treated cells than untreated cells (*Figure 5A*). This result was confirmed in wild-type cells (*Figure 5—figure supplement 1B*), indicating it was not the result of the *spo11-Y135F* allele. Spo11 levels were partially restored in *mec1Δ* and *cdc6-mn* mutants, but not *rad53Δ* cells (*Figure 5A*), indicating that downregulation of Spo11 protein is a result of Rad53-independent replication checkpoint activity. Northern blot analysis indicated that checkpoint-dependent downregulation of Spo11 occurs at the level of the *SPO11* transcript (*Figure 5B,C*, *Figure 5—figure supplement 1C*). *SPO11* RNA accumulation halted 1 hr after meiotic induction (the time of replication onset), ultimately leading to a reduction in steady-state RNA levels of at least 5-fold (*Figure 5C*, *Figure 5—figure supplement 1C*). We note that *SPO11* RNA and protein were less abundant in *rad53Δ* cells than wild type, which may contribute the

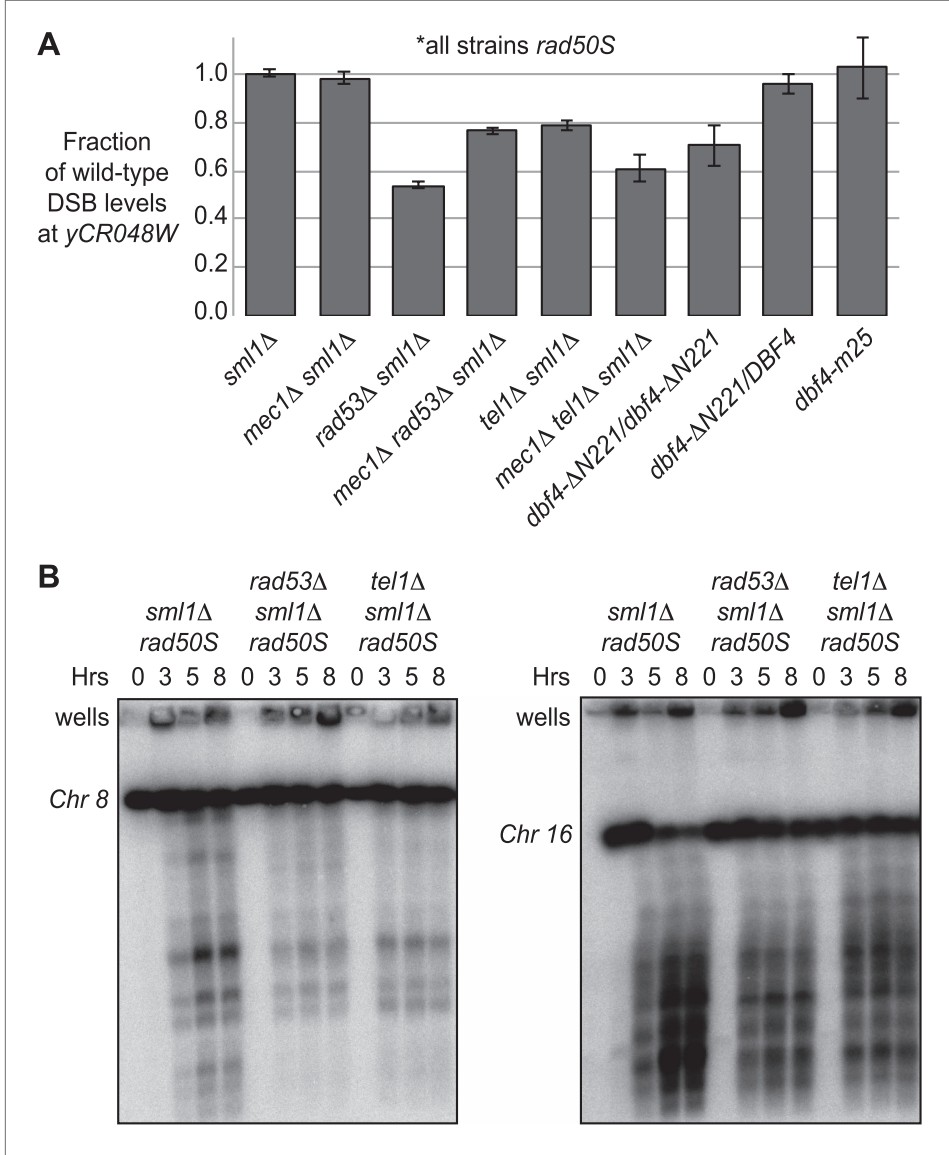

**Figure 4**. A Tel1-dependent feedback mechanism increases DSB levels in *rad50S* cells. (**A**) The maximum levels of DSBs in untreated *rad50S* strains containing the indicated mutations were measured from the Southern blots shown in *Figure 2*, *Figure 3*, *Figure 2—figure supplement 1*, *Figure 3—figure supplement 1*, and *Figure 4—figure supplement 1*. The amount of broken DNA was calculated as in *Figure 1* and all values were normalized to the wild-type strain from the same experiment. (**B**) CHEF gel electrophoresis and Southern blotting was conducted to assess DSB levels on whole chromosomes in *sml1Δ rad50S* (H4898), *rad53Δ sml1Δ rad50S* (H4969), and *tel1Δ sml1Δ rad50S* (H4849) cells. Chromosomes 8 (left panel) and 16 (right panel) were resolved on separate gels, blotted and visualized with the following probes in SGD coordinates: Chromosome VIII: 23,771-25,410 and Chromosome XVI: 20,281-21,012. See also *Figure 4—figure supplement 1*.

The following figure supplements are available for figure 4:

**Figure supplement 1**. *TEL1* is required for wild-type DSB levels in *rad50S* cells.

decreased DSBs levels in this strain background. We conclude that *SPO11* expression is under checkpoint control.

Our Western blot analysis also revealed that several DSB factors exhibited altered electrophoretic mobility upon HU treatment, indicative of a possible altered phosphorylation state. These changes are

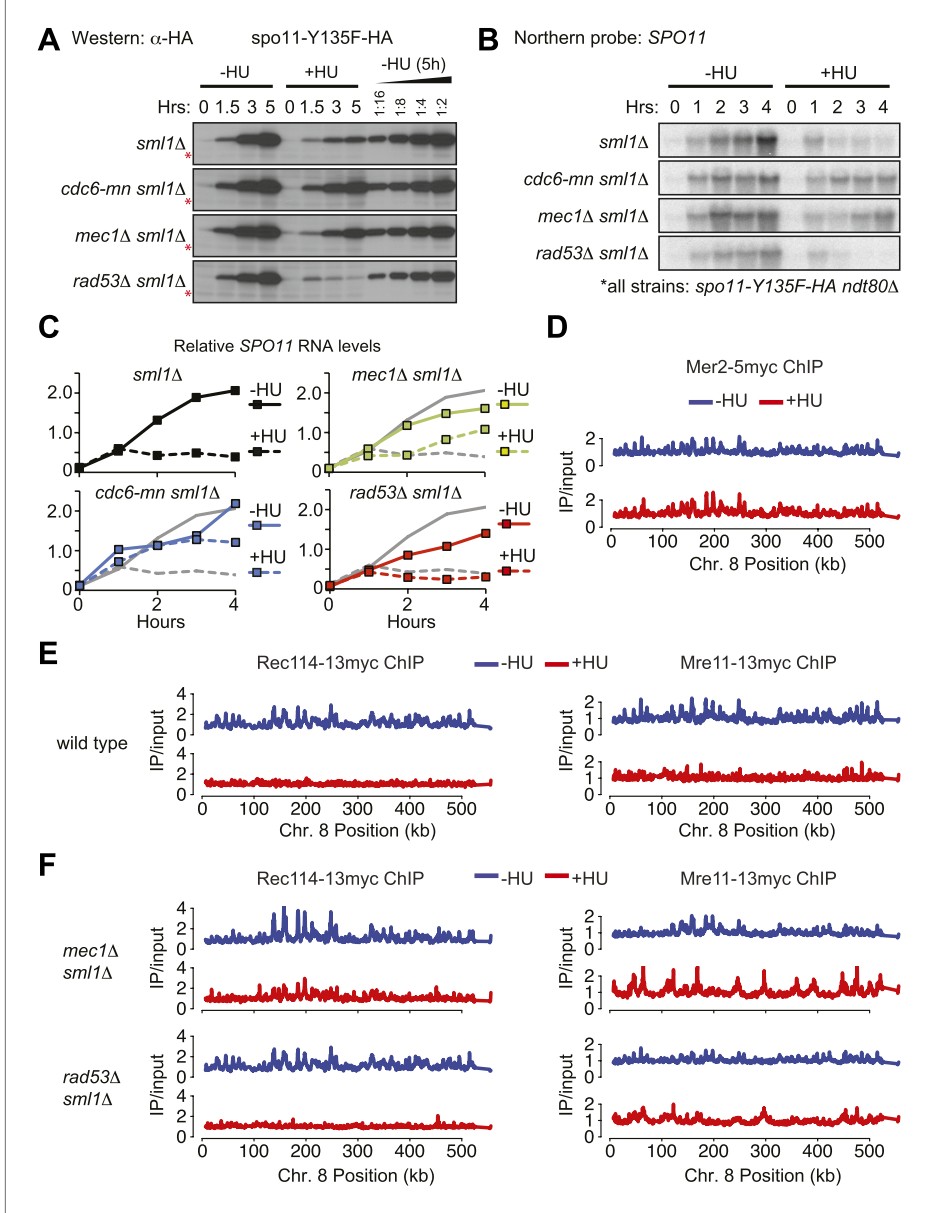

**Figure 5**. The replication checkpoint regulates DSB factor levels and DNA loading. (**A–C**) Analysis of *spo11-Y135F-HA* in *sml1Δ ndt80Δ* (H5233), *cdc6-mn sml1Δ ndt80Δ* (H7447), *mec1-1 sml1Δ ndt80Δ* (H5227), *rad53Δ sml1Δ ndt80Δ* (H5230) strains in the presence or absence of HU. (**A**) Western blot analysis of spo11-Y135F-HA levels. A twofold dilution series of the 5h (–HU) time point was used to estimate changes in protein levels in the presence of HU. The cross-reacting band marked with red asterisks serves as a loading control. (**B**) Northern blot analysis of *spo11-Y135F-HA* RNA. (**C**) Quantification of the Northern blots in (**B**). Northern blots were reprobed for *UBC6* (***Teste et al., 2009***) for normalization. RNA levels of the *spo11-Y135F-HA sml1Δ ndt80Δ* 'wild-type' strain are shown as grey lines in all panels for comparison. (**D**) Binding profiles for Mer2-13myc (H4585) from genome-wide location analysis (ChIP-chip) along Chromosome 8 in the absence (blue lines) or presence (red lines) of HU. (**E**) As in (**D**), binding profiles from ChIP-chip analysis for Rec114-13myc (H4890) and Mre11-13myc (H5547) in wild-type cells in the absence (blue lines, ***Vader et al., 2011***) or presence (red lines) of HU. (**F**) ChIP-chip binding profiles of and Rec114-13myc in *mec1Δ sml1Δ* (H7305) and *rad53Δ sml1Δ* (H7302) cells and for Mre11-13myc in *mec1Δ sml1Δ* (H7323) and *rad53Δ sml1Δ* (H7320) cells in the absence (blue lines) or presence (red lines) of HU. See also ***Figure 5—figure supplement 1***.

The following figure supplements are available for figure 5:

**Figure supplement 1**. The meiotic replication checkpoint regulates DSB factor abundance, phosphorylation and DNA binding.

unlikely to be a consequence of direct phosphorylation by Mec1, as no protein showed the expected reduced mobility in HU-treated cells (*Figure 5—figure supplement 1A*). However, we noted that several protein bands showed faster mobility upon HU treatment, including Mer2, Rec104 (*Figure 5B*), Sae2/CtIP and Rec8 (*Figure 5—figure supplement 1A*). As Mer2 and Rec8 are both reported to be DDK targets, it is possible that the increased mobility of these proteins in the presence of HU is the result of the decreased DDK activity we observed. Sae2 shows checkpoint- and cell-cycle dependent phosphorylation (*Huertas et al., 2008*; *Manfrini et al., 2010*) and Rec104 has previously been described as a meiotic phospho-protein (*Kee et al., 2004*). These data raise the intriguing possibility that multiple DSB and repair factors are targets of DDK-dependent phosphorylation, including Rec104 and Sae2.

To test whether the meiotic replication checkpoint affected the localization of DSB factors, we analyzed their DNA binding by genome-wide location analysis in the absence and presence of HU. Mer2-5myc was similarly detected on the same core meiotic chromosome binding sites occupied by axial proteins (*Panizza et al., 2011*) in the absence or presence of HU (*Figure 5D*), in spite of the fact that it is not fully phosphorylated in HU-treated cells (*Figure 2A*, *Figure 5—figure supplement 1A*). In contrast, we were unable to detect chromosomal loading of Rec114-13myc and Mre11-13myc in HU-treated cells, although both proteins associated with the DNA robustly in the absence of HU (*Figure 5E*). These data suggest that the loading of specific DSB factors is prevented by the replication checkpoint.

To understand how the replication checkpoint prevents Rec114 and Mre11 DNA loading, we monitored their chromosomal association in *mec1Δ* and *rad53Δ* cells. We found that deletion of *MEC1* restored Rec114-13myc loading in the presence of HU whereas deletion of *RAD53* did not (*Figure 5F*, left panels), indicating that inhibition of Rec114 chromosome loading is Mec1-specific. Given that Rec114 forms a complex with Mer2 (*Arora et al., 2004*), it is possible that Rec114 DNA loading depends on Mer2 phosphorylation. However, we do not believe this to be case, as Mer2 was equally phosphorylated in both *mec1Δ* and *rad53Δ* cells (*Figure 2A*), yet Rec114 loading was specifically regulated by Mec1. Furthermore, Rec114-13myc could not be detected on chromosomes in HU-treated cells containing the *mer2-DDD* allele that mimics the DDK-dependent phosphorylations (*Wan et al., 2008*) (data not shown), consistent with the idea that Mer2 phosphorylation is not sufficient to recruit Rec114 to the DNA when Mec1 is activated.

Similar to our results for Rec114, we found that chromosome loading of Mre11 was detectible in HU-treated *mec1Δ*, but not *rad53Δ* cells (*Figure 5F*, right panels). However, the pattern of Mre11 DNA-binding was altered dramatically in checkpoint mutant cells (*mec1Δ* and *rad53Δ*) treated with HU; Mre11 association with core meiotic chromosomal binding sites was substantially lower than in untreated cells, and the protein instead accumulated close to every pre-meiotic replication origin (*Blitzblau et al., 2012*) (*Figure 5—figure supplement 1D*, dashed grey lines). Indeed, Mre11 was significantly enriched specifically at replication origins in *mec1Δ* cells treated with HU (*Figure 5—figure supplement 1E*, p values from Student's *t* test). We believe this to be the result of Mre11 recruitment to the aberrant DNA damage structures that form at replication forks when the replication checkpoint is impaired (*Feng et al., 2006*). This extensive relocalization of Mre11 may explain why DSB levels in *mec1Δ* cells are lower in the presence of HU than without treatment (*Figure 3*). The specific inhibition of Rec114 and Mre11 loading by Mec1 reveals an additional Rad53-independent replication checkpoint pathway.

## DSBs on replicating chromosomes result in lethal genome fragmentation

The presence of redundant mechanisms to inhibit meiotic DSB factors in response to DNA replication suggests that there are negative consequences of DSBs on replicating chromosomes. We were unable to test this hypothesis in the *mec1Δ* strain, as these cells die due to replication fork problems upon HU treatment (*Figure 1—figure supplement 1A*). Therefore, we employed the *cdc6-mn* strain, which experiences high levels of meiotic DSBs on chromosomes that have undergone very little replication (schematic *Figure 6A*). Although Cdc6 protein levels are strongly depleted in *cdc6-mn* mutants (data not shown), low levels of DNA replication were observed in this strain (*Figure 6B*, shoulder to the right of 2C DNA). To rule out the possibility that this was break-induced replication (BIR), we depleted DSBs in the *cdc6-mn* strain using a hypomorphic *SPO11* allele. Remarkably, rather than observing a reduction in DNA replication, as would be expected if the shoulder in the FACS profile was due to BIR, we observed

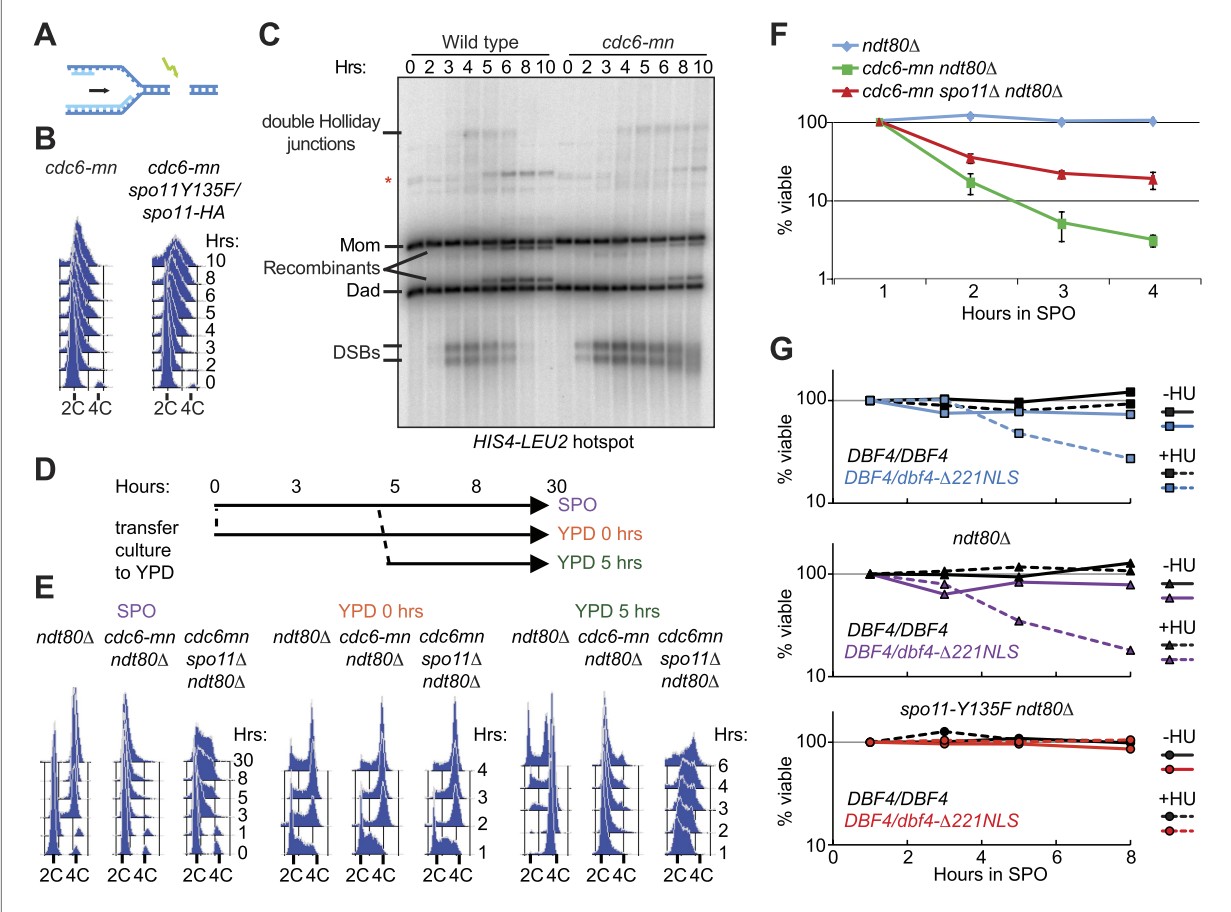

**Figure 6**. DSBs on replicating chromosomes are lethal. (**A**) Schematic of a DSB that occurs ahead of a DNA replication fork. (**B**) FACS analysis of total DNA content in *cdc6-mn* (H2655) and *cdc6mn spo11-HA3-His6/spo11-Y135F* (H3598) cells as they progress through meiosis. (**C**) Southern blot analysis of DSB formation and repair at the *HIS4-LEU2* hotspot in wild-type (H2636) and *cdc6-mn* (H2655) cells. Cells were treated with psoralen and DNA was crosslinked with UV light to preserve recombination intermediates. The relative positions of parental bands, repair intermediates and recombinants are marked. The red asterisk marks the position of an alternative recombination product. (**D**) Schematic of the experiment shown in (**E** and **F**). Cells from *ntd80Δ* (H385), *cdc6-mn ndt80Δ* (H386) and *cdc6-mn spo11Δ ndt80Δ* (H3682) pre-sporulation cultures were split into YPD or SPO to induce mitosis or meiosis, respectively. After 5 hr, half of the SPO culture was returned to growth in YPD. (**E**) FACS analysis of total DNA content was performed for the experiment described in (**D**). (**F**) Viability of cells was measured for the strains described in (**D**) at the indicated time points during meiotic induction. The number of colonies at each time point was normalized to the 1-hr time point for each culture. (**G**) Viability of wild-type (H7099), *dbf4-NLS-ΔN221/DBF4* (7401), *ndt80Δ* (H7494), *dbf4-NLS-ΔN221/DBF4 ndt80Δ* (H7493), *ndt80Δ spo11-Y135F-HA* (H7468) and *dbf4-NLS-ΔN221/DBF4 ndt80Δ spo11-Y135F-HA* (H7469) strains induced to enter meiosis in the presence or absence of HU and transferred onto YPD medium at the indicated time points. Each point is the average of two or three independent experiments. Viabilities were normalized to the 1-hr time point for each culture. See also ***Figure 6—figure supplement 1***.

The following figure supplements are available for figure 6:

**Figure supplement 1**. Characterization of DNA replication and SC formation in *cdc6-mn* cells.

a substantial increase in DNA replication (***Figure 6B***). This suggests that DSBs in the *cdc6-mn* strain inhibit DNA replication, possibly by directly blocking replication forks and/or activating a DNA damage checkpoint.

We investigated the source of the DNA replication we observed in *cdc6-mn* cells. The DNA content increase was not due to mitochondrial DNA replication (***Figure 6—figure supplement 1A***), but rather because whole chromosomes were sporadically replicated and segregated in the *cdc6-mn* strain, as revealed by the occasional duplication and segregation of a single chromosome marked with a *TetO* array/TetR-GFP (***Figure 6—figure supplement 1B***). Although we were unable to detect the loading of Mcm2-7 helicase in *cdc6-mn* cells by genome-wide location analysis (***Figure 1—figure supplement 1B***),

we believe that the *SCC1* promoter driving *CDC6* expression in the *cdc6-mn* strain allows for leaky expression that permits an undetectable amount of Cdc6 to act stochastically at replication origins throughout the genome. We suggest that the infrequency of these events precludes detection by population-based assays. Consistent with this idea, when we measured DNA replication in the *cdc6-mn* cells after 8 hr in sporulation medium compared to G1 unreplicated DNA, no specific chromosomes or regions of the genome were preferentially replicated except the ribosomal DNA (rDNA) (*Figure 6—figure supplement 1C*). Because the rDNA contains ~100 tandem 9.1 kb repeats, each with its own potential origin of replication, the chance that one of these repeats would load Mcm2-7 and initiate DNA replication is extremely high, compared to the 11–46 origins present in the single-copy regions of each chromosome. Therefore, we conclude that *cdc6-mn* cells undergo sporadic chromosomal replication initiation, although clearly the levels of replication are too low to promote a checkpoint response in the critical period, during which DSBs must be prevented (*Figure 1H*).

To understand the consequences of forming DSBs on replicating chromosomes, we analyzed DSB repair in *cdc6-mn* cells at the well-characterized *HIS4-LEU2* locus on chromosome 3 (*Hunter and Kleckner, 2001*). Engineered restriction sites at this locus permit the measurement of DSB repair by Southern blot analysis. In wild-type cells, DSB formation is followed by the accumulation of double-Holliday junction repair intermediates. These intermediates are subsequently resolved into crossover products that show an altered molecular weight from the parental fragments (*Figure 6C*). DSBs formed at wild-type levels at *HIS4-LEU2* in *cdc6-mn* cells, and a subset of DSBs were converted into crossovers, indicating that *cdc6-mn* cells are able to undergo homologous recombination. However, we also observed many DSBs that accumulated and migrated faster in the gel at 8–10 hr, indicating that repair of these breaks is defective and they become hyperresected (*Figure 6C*). Similarly, *cdc6-mn* cells exhibited a small defect in synaptonemal complex formation between homologous chromosomes (*Figure 6—figure supplement 1D,E*), likely a result of the repair defects we observed. Thus, a subset of DSBs persist in *cdc6-mn* mutants, indicating that cells that form DSBs on partially replicated chromosomes are unable to complete DSB repair.

Given that *cdc6-mn* cells have problems completing both DNA replication and DSB repair, we asked whether these defects affected later meiotic events or cell viability. Spindle and DAPI analysis revealed that despite ongoing DNA replication and unrepaired DSBs, *cdc6-mn* cells entered into the meiotic divisions with little delay from the wild-type cells (*Hochwagen et al., 2005*), but exhibited strongly reduced levels of tetranucleate formation (*Figure 6—figure supplement 1C*) and no viable spores were produced (data not shown). This result suggests that DSBs formed during meiS are catastrophic for meiotic cells, which are unable to complete DNA replication or restrain the nuclear divisions.

We employed the return-to-growth protocol to determine the contribution of precocious DSB formation to cell lethality. DSB repair in meiotic cells is constrained to promote homologous recombination. Some DSBs, for example in *dmc1Δ* cells, cannot be repaired during meiosis, but can be repaired if cells are returned to mitotic growth in rich medium (*Shinohara et al., 1997*; *Zenvirth et al., 1997*). We analyzed *ndt80Δ*, *cdc6-mn ndt80Δ* and *cdc6-mn spo11Δ ndt80Δ* strains that were blocked in meiotic prophase to prevent meiotic chromosome segregation (experimental outline in *Figure 6D*), and measured DNA replication (*Figure 6E*) and viability (*Figure 6F*). Cells returned to rich medium prior to meiotic entry (0 hr), should express *CDC6* normally from the *SCC1* promoter, and, accordingly, we observed no defect in DNA replication or cell division (*Figure 6E*, middle three panels). When cells remained in sporulation medium, little DNA replication occurred in *cdc6-mn ndt80Δ* cells, but much more was observed in *cdc6-mn spo11Δ ndt80Δ* cells by 30 hr (*Figure 6E*, left three panels), showing that DSBs substantially impeded the completion of DNA replication in the *cdc6-mn* background. When cells were returned to growth in rich medium after 5 hr in sporulation medium, wild-type cells had already completed DNA replication (*Figure 6E*, first right panel), and returned to cycling with 100% viability (*Figure 6F*). In contrast, *cdc6-mn* cells were able to complete very little DNA replication upon return to growth (*Figure 6E*, second to last panel) and lost viability quickly after exposure to sporulation medium (*Figure 6F*). Removal of Spo11 allowed more DNA replication upon return to growth (*Figure 6E*, compare last two panels) and the *spo11Δ* strain exhibited significantly increased viability over the *SPO11* strain. We confirmed that replication-checkpoint bypass similarly results in HU- and Spo11-dependent lethality in the *dbf4-NLSΔN221* strain background in unblocked meiosis (top panel), as well as in prophase-arrested cells (*ndt80Δ*) (*Figure 6G*). Together, these results reveal that preventing DSB formation on replicating chromosomes is essential to inhibit a lethal meiotic chromosome fragmentation event that significantly impedes both the completion of DNA replication, and the repair of meiotic DSBs.

## Discussion

The data presented here demonstrate that the replication checkpoint functions during budding yeast meiosis to maintain genome stability and viability. The checkpoint employs multiple strategies to inhibit DSB factors while DNA replication is ongoing. Without this restraining mechanism, cells form high levels of DSBs on partially replicated chromosomes, and lose the ability to complete either DNA replication or DSB repair.

### A conserved pre-meiotic replication checkpoint

In this study we have defined the core components of a pre-meiotic replication checkpoint in budding yeast. Meiotic cells detect replication stress using the canonical replication checkpoint machinery, which then regulates meiosis-specific processes. As during mitotic S phase, detection of HU-induced replication inhibition relies on Mec1 and Ddc2. Mec1 activates the Rad53 effector kinase, which, in turn, inhibits DDK activity. The essential roles of Mec1 and Rad53 in maintaining replication fork stability appear to be equally important in meiotic cells, as removing either protein is lethal when cells are exposed to HU in meiS. Similar to the mitotic replication checkpoint, redundant and separable Mec1- and Rad53-dependent mechanisms maintain genome stability and prevent precocious cell-cycle progression in the presence of replication inhibition during meiosis (*Labib and De Piccoli, 2011*). However, the targets of the checkpoint differ.

We found that the pre-meiotic replication checkpoint prevents accumulation, DNA-loading and phosphorylation of DSB factors (*Figure 7*, model). First, Mec1 downregulates *SPO11* transcript levels. Although Spo11 production is not completely prevented, we believe that downregulation of Spo11 has functional consequences because genetically decreasing Spo11 activity was able to significantly

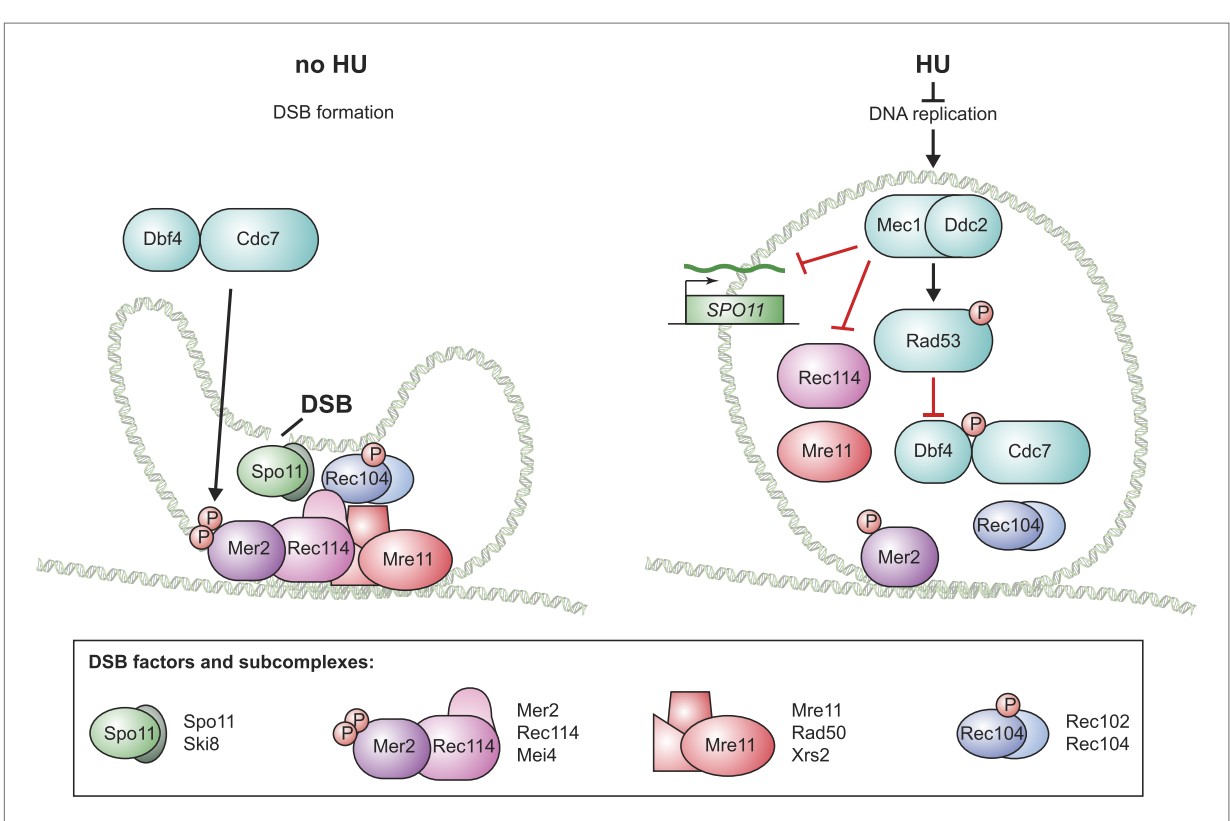

**Figure 7**. Model for the pre-meiotic replication checkpoint in budding yeast. A schematic for the assembly of the four budding yeast DSB factor complexes (as defined below) in the absence (left) and presence (right) of replication inhibition is shown. In the absence of inhibition, all factors load onto the DNA and Mer2 and Rec104 are fully phosphorylated, allowing Spo11 to introduce DSBs. In the presence of HU, the levels of *SPO11* transcripts are reduced, the DNA loading of Mre11 and Rec114 is prevented and the phosphorylation of Mer2 and Rec104 is abrogated. Illustration by Tom DiCesare (Whitehead Institute).

rescue DNA replication in the *cdc6-mn* strain (*Figure 6B*). Second, Mec1 activity strongly reduces the DNA loading of Rec114 and Mre11. Rec114 is phosphorylated in a Mec1-dependent manner in response to meiotic DSBs (*Carballo et al., 2013*). However, such phosphorylation was found to increase Rec114 chromosomal association and we found no evidence that Rec114 or Mre11 are direct targets of Mec1 during meiS. The lack of DSB factor loading could be due to impaired axis formation, in spite of the fact that Rec8, Hop1 and Red1 load robustly onto the DNA in HU-treated cells (*Blitzblau et al., 2012*). Alternatively, the DSB factors may be directly regulated by the replication checkpoint. Third, Rad53-dependent inhibition of DDK prevents Mer2 phosphorylation, which is expected to block meiotic DSBs (*Sasanuma et al., 2008*; *Wan et al., 2008*). The redundancy resulting from the separable inhibition of all of the major DSB complexes likely increases the speed and robustness of the checkpoint response, which is important given the severe repair defects and lethality associated with precocious DSB formation on unreplicated DNA.

The role of Dbf4 in the replication checkpoint remains elusive. Studies in pre-mitotic cells indicate that DDK activity is regulated by Rad53-dependent phosphorylation and interaction (*Zegerman and Diffley, 2010*; *Chen et al., 2013*), but our results separate the phosphorylation status and activity of the protein from the ability to form DSBs in HU-treated cells. HU treatment resulted in Mec1- and Rad53-dependent Dbf4 phosphorylation and a block of DDK target phosphorylation. However, neither removal of Rad53 (which prevented Dbf4 hyperphosphorylation and allowed for Mer2 phosphorylation) nor the dbf4-m25 non-phosphorylatable protein allowed DSBs in the presence of HU, indicating that simply allowing DDK activation does not bypass the pre-meiotic replication checkpoint. Furthermore, the dbf4-Δ71-221 protein that cannot interact with Rad53 (*Chen et al., 2013*) also did not allow for checkpoint bypass. The only form of Dbf4 that allowed for efficient DSB formation in HU-treated cells was dbf4-NLSΔN221, which appears to be an unregulated version of the protein that is dominant to the wild-type. The fact that the *dbf4-NLSΔN221* strain is not sensitive to HU in mitotic cells (*Gabrielse et al., 2006*) suggests that this allele does not bypass checkpoint activation, but rather acts downstream or in parallel to Mec1 and Rad53. It is possible that dbf4-NLSΔN221 is hyperactive compared to the wild-type protein and overcomes the checkpoint, or that it accumulates DDK activity early and 'licenses' DSBs before the replication checkpoint is activated. Regardless of the mechanism, our results clearly indicate that controlling DDK activity is a critical step in regulating DSB formation.

## Replication checkpoint activation and specificity

It is intriguing that there is so little Rad53-dependent checkpoint activation in *cdc6-mn* cells, in spite of the observation that they can initiate significant DNA replication, as evidenced in *spo11* mutant cells. One possibility is that DSB formation in *cdc6-mn* cells triggers the activation of the recombination checkpoint, in which Mec1 specifically activates the Mek1 kinase and not Rad53 (*Usui et al., 2001*; *Cartagena-Lirola et al., 2008*) to direct DSB repair activity to the homologous chromosomes (*Carballo et al., 2008*; *Niu et al., 2009*). Therefore, Rad53 activity may be specifically suppressed during prophase by the DSBs in *cdc6-mn* cells. Rad53 can be activated later in meiosis (*Cartagena-Lirola et al., 2008*), and consistently, we noted that Rad53 became activated at later time points in *cdc6-mn* cells treated with HU (*Figure 1H*), which proceed into the meiotic divisions with replication forks and DSBs. An alternative and non-exclusive model is that no replication checkpoint is activated in *cdc6-mn* cells either because Cdc6 itself is a checkpoint factor, as has been suggested for *S. pombe* (*Hermand and Nurse, 2007*), or the number of replication forks is too low to promote checkpoint activation. In either case, the observed restriction of Rad53 activity could allow cells to treat DNA lesions differently depending on the type of damage and phase of the cell cycle.

## Coordinating DNA replication with DSB formation

The pre-meiotic replication checkpoint delays DSB formation and cell cycle progression during impaired DNA replication, but it is not the only mechanism to preserve cell cycle order. Measuring the kinetics of DNA replication and DSB formation in both unchallenged and HU-treated cells indicated that DSB formation occurs with a fixed delay with respect to the appearance of 4C (replicated) DNA, consistent with previous observations (*Padmore et al., 1991*). Therefore, it appears that under normal circumstances, DSBs occur only after bulk DNA replication. Furthermore, DSB formation seems to be timed independently from DNA replication, as it did not occur significantly earlier in *cdc6-mn* replication-depleted or checkpoint mutants cells, suggesting the order and timing of the two processes are under

independent cell cycle controls. A second mechanism to directly couple DNA replication and DSB formation has been described; delaying DNA replication on the left arm of chromosome 3 similarly delays local DSB formation (*Borde et al., 2000*). However, we do not believe this coordination is the same as the checkpoint described here, as delaying DNA replication with low concentrations of HU was sufficient to block DSB formation until 4C DNA appeared, that is when meiS was completed. The local delay of DSB formation with late forks is likely necessary, because we have shown here that the replication checkpoint is insensitive to very low levels of DNA replication, such as in the *cdc6-mn* mutant. Therefore, any genomic regions with late forks that do not complete replication inside of the normal S phase would be subject to this second coupling mechanism to ensure DSBs do not occur prior to replication completion. The severe phenotype exhibited by cells that make breaks on replicating chromosomes could explain the existence of multiple coordinating mechanisms for meiS and DSB formation.

### Conservation of the pre-meiotic replication checkpoint

A pre-meiotic replication checkpoint that prevents DSBs in response to replication inhibition is conserved in the distantly related fission yeast *S. pombe*, where Rad3/ATR/Mec1 and Cds1/CHK2/Rad53 are similarly activated in response to HU treatment. Some mechanistic differences exist, which could be due to the lack of identity of the DSB factors themselves. First, unlike budding yeast, fission yeast lacking the Rad53 homolog Cds1 form DSBs in the presence of HU (*Tonami et al., 2005*; *Ogino and Masai, 2006*). Second, although specific DSB factors are transcriptionally downregulated in *S. pombe* (*Ogino and Masai, 2006*; *Miyoshi et al., 2012*), their identity is not conserved in budding yeast and the *S. pombe* Spo11-homolog Rec12 was not affected. Other mechanistic parallels remain to be tested, in particular, the role of DDK in the pre-meiotic replication checkpoint, which we identified as a central player in coupling DNA replication with DSB formation in budding yeast. The *S. pombe* Cdc7 homolog Hsk1 is essential for DSB formation (*Ogino et al., 2006*), and is regulated by the pre-mitotic replication checkpoint (*Snaith et al., 2000*). The close temporal succession of pre-meiotic DNA replication and DSB formation is a universal feature of meiotic recombination, and our data demonstrate that the separation of these two processes is vital for maintaining genomic integrity. Given that both the DDK cell cycle kinase and the replication checkpoint are highly conserved in all eukaryotes, it seems likely that similar coupling mechanisms exist to protect the gametes in other species, including humans.

## Materials and methods

### Strains and growth conditions

Strains used in this study are isogenic to SK1 and are listed in *Supplementary file 1*. Gene disruptions and tagging were carried out using a PCR-based protocol (*Longtine et al., 1998*). Synchronous meiosis was induced as previously described (*Blitzblau et al., 2012*). For HU experiments, cells were inoculated into sporulation medium (SPO) containing 20 mM HU at 30°C, except in *Figure 1* when 5 mM HU was used as indicated.

### Yeast viability assays

The proportion of viable cells in the culture was measured at each indicated time point by removing and plating ~500 cells on YPD plates and measuring the number of colonies that grew after 3 days at 30°C. The number of colonies present was normalized to the number observed at the first time point (0 or 1 hr after introduction into SPO).

### FACS analysis

FACS analysis for total DNA content was performed as in (*Blitzblau et al., 2012*).

### CHEF and Southern analysis

Clamped-homogeneous electric field (CHEF) gel electrophoresis and Southern blotting for small chromosomes (including chromosome 8) were performed as described (*Blitzblau et al., 2007*). Large chromosome CHEF analysis was carried out similarly, using a 1% gel for 15 hr with 60 s pulses followed by 9 hr at 90 s. For resolution of recombination intermediates, cells were killed with 0.1% sodium azide. They were resuspended in 0.1 mg/ml psoralen in TE (50 mM Tris pH 7.5, 50 mM EDTA) and crosslinked with 365-nm UV light for 12 min on a UV lightbox (5 mW/cm$^2$) in a polystyrene culture dish.

DNA was isolated via standard Southern protocol. All time course experiments for Southern analysis were repeated at least twice with similar results.

## Western blot analysis

Whole cell protein extracts were prepared by TCA precipitation as in *Blitzblau et al. (2012)*. An equal number of cells were loaded for each sample and equivalent loading was confirmed by Ponceau S staining. SDS-polyacrylamide gel electrophoresis and blotting were performed as described in *Falk et al. (2010)*. The following antibodies were used for detections, all diluted in PBS-T + 3% milk (TBS-T was used for Mer2 phospho-S30) and incubated overnight; anti-Rad53 yC-19 (Santa Cruz Biotechnology Inc, Santa Cruz, CA) used at 1:5,000 dilution and Rad53 separated on an 8% gel; rabbit polyclonal anti-Dbf4 HM5765 (Steven P Bell, *Francis, et al., 2009*) was used at 1:1,000 dilution and Dbf4 separated on a 7.5% gel; anti-myc 9E10 (Covance, Princeton, NJ) was used at 1:1,000 for the following proteins: Mer2, Rec104, and Sae2 separated on 14% gels, Rec114, Mei4, and Rec102 separated on 12% gels, Spo11, Mre11, Xrs2, and Ski8 separated on 10% gels; anti-HA 12CA5 (Roche, Basel, Switzerland) was used at a 1:1,000 dilution for Rad50 separated on a 7.5% gel and Rec8 separated on a 10% gel; anti-HA 3F10 (Roche) was used at 1:1,000 for spo11-Y135F-HA separated on a 12.5% gel; anti-Mer2 phospho-S30 (Abcam, Cambridge, MA) was used at a 1:1,000 dilution. The appropriate species secondary antibodies were diluted 1:5,000 in PBS-T + 3% milk (except TBS-T was used for Mer2 phospho-S30) and incubated for 2 hr at room temperature.

## Northern blot analysis

Northern blot analysis was performed as in *Hochwagen et al. (2005)* with minor modifications. 6 ml of cells were harvested at the indicated time points, washed once with 1 × TE (10 mM Tris pH 7.5, 1 mM EDTA) and frozen at −80°C. Cell pellets were ruptured by vigorous shaking for 30 min at 4°C in equal volumes glass beads, phenol-chloroform-isoamylalcohol (25:24:1) and cold RNA buffer 1 (300 mM NaCl, 10 mM Tris pH 7.5, 1 mM EDTA, 0.2% SDS). RNA was precipitated in ethanol and resuspended at 65°C in RNA buffer 2 (10 mM Tris pH 7.5, 1 mM EDTA, 0.2% SDS). 30 μg of each RNA sample were denatured at 65°C in denaturing solution (50% formamide, 6.5% formaldehyde, 40 mM MOPS pH 7.0, 10 mM sodium acetate, 0.1 mM EDTA) and separated by electrophoresis in a 1.1% agarose gel (containing 6% formaldehyde) in MOPS buffer (40 mM MOPS pH 7.0, 10 mM sodium acetate, 0.1 mM EDTA). RNA was blotted onto a Zeta probe GT membrane (BioRad) in 10 × SSC (1.5 M NaCl, 150 mM sodium citrate, pH 7.0) and UV-crosslinked using a Stratalinker. Probe hybridization was performed as for Southern blots.

## Chromatin immunoprecipitation

25 ml of cells were harvested after 3 hr in SPO. Chromatin immunoprecipitation (ChIP) for genome-wide location analysis was performed as described (*Blitzblau et al., 2012*). One tenth of the lysate was removed as an input sample. Samples were immunoprecipitated for 16 hr at 4°C with anti-HA 3F10 (Rec8-3HA, Roche, used 2 μg per immunoprecipitation), ChIP grade anti-myc 9E11 (Abcam, 2 μl used per immunoprecipitation) or UM185 (Rabbit polyclonal anti-Mcm2-7, Stephen P Bell, 2 μl serum used per immunoprecipitation).

## DNA extractions for CGH

Total genomic DNA extraction for CGH analysis was performed as described in *Blitzblau et al. (2012)*.

## Fluorescent labeling and microarray hybridization

For ChIP experiments, one half of the immunoprecipitated DNA and one tenth of the input DNA were labeled. Samples were labeled and hybridized as in *Blitzblau et al. (2012)*.

## Microarray data analysis

For each co-hybridization, Cy3 and Cy5 levels were calculated using Agilent Feature Extractor CGH software. Background normalization, $\log_2$ ratios for each experiment and scale normalizations across each set of biological replicates were calculated using the sma package (*Yang et al., 2001*) in R, a computer language and environment for statistical computing (v2.1.0, http://www.r-project.org). The value of the feature closest to each potential pre-meiotic replication origin (*Blitzblau et al., 2012*) was used to estimate Mre11 binding close to replication origins. The raw data and log ratios analyzed in this study are available from the NCBI Gene Expression Omnibus (http://www.ncbi.nlm.nih.gov/geo/), accession number GSE46841.

## Indirect immunofluorescence on spread nuclei

Meiotic nuclear spreads were performed as described (*Falk et al., 2010*). In brief, the nuclei of spheroplasted cells were spread on a glass slide in the presence of paraformaldehyde fixative and 1% lipsol. After drying, the slides were blocked in blocking buffer (0.2% gelatin, 0.5% BSA in PBS) and stained with anti-Rad51 y-180 (Santa Cruz) used at 1:200 dilution and anti-Zip1 yN-16 (Santa Cruz) used at 1:100 dilution.

## Acknowledgements

We would like to thank Steve Bell for Mcm2-7 and Dbf4 antibodies; David Toczyski, Nancy Hollingsworth, Scott Keeney, John Petrini, and Michael Weinreich for plasmids; and Gerben Vader, Steve Bell, and Hannah Klein for comments on the manuscript. We especially thank Gerry Fink for generously providing lab space to HB and critical feedback on the project.

## Additional information

### Funding

| Funder | Grant reference number | Author |
| --- | --- | --- |
| Charles A King Trust, Bank of America | | Hannah G Blitzblau |
| Alexander and Margaret Stewart Trust Cancer Pilot Grant | | Andreas Hochwagen |
| National Institutes of Health | R01 GM088248 | Andreas Hochwagen |

The funders had no role in study design, data collection and interpretation, or the decision to submit the work for publication.

### Author contributions

HGB, AH, Conception and design, Acquisition of data, Analysis and interpretation of data, Drafting or revising the article

## Additional files

### Supplementary files

• Supplementary file 1. Genotypes of yeast strains used in this study.

### Major dataset

The following dataset was generated:

| Author(s) | Year | Dataset title | Dataset ID and/or URL | Database, license, and accessibility information |
| --- | --- | --- | --- | --- |
| Blitzblau HG, Hochwagen A | 2013 | Data from: DNA replication and DSB factor binding in the meiotic S-phase checkpoint in budding yeast | GSE46841; http://www.ncbi.nlm.nih.gov/geo/query/acc.cgi?acc=GSE46841 | Publicly available at the Gene Expression Omnibus (http://www.ncbi.nlm.nih.gov/geo/). |

The following previously published datasets were used:

| Author(s) | Year | Dataset title | Dataset ID and/or URL | Database, license, and accessibility information |
| --- | --- | --- | --- | --- |
| Blitzblau HG, Chan CS, Hochwagen A, Bell SP | 2012 | Data from: Pre-meiotic Mcm2-7 ChIP | GSE35667; http://www.ncbi.nlm.nih.gov/geo/query/acc.cgi?acc=GSE35667 | Publicly available at the Gene Expression Omnibus (http://www.ncbi.nlm.nih.gov/geo/). |

| Vader G, Blitzblau HG, Tame MA, Falk JE, Curtin L, Hochwagen A | 2011 | Data from: ChIP-chip of DSB factors in wild type and pch2 strains | GSE30072; http://www.ncbi.nlm.nih.gov/geo/query/acc.cgi?acc=GSE30072 | Publicly available at the Gene Expression Omnibus (http://www.ncbi.nlm.nih.gov/geo/). |

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
