## [Decision Letter]

Thank you for sending your work entitled “ATR/Mec1 prevents lethal meiotic recombination initiation on partially replicated chromosomes in budding yeast” for consideration at *eLife*. Your article has been evaluated by a Senior editor and 4 reviewers, one of whom is a member of our Board of Reviewing Editors. Although there was some disagreement among reviewers over the novelty of the findings, on balance they came to the conclusion that if you can address the comments enumerated below, the paper will represent a valuable contribution to the field that merits publication in *eLife*. If you are not able to address the key comments, we do not advise resubmission.

The Reviewing editor and the other reviewers discussed their comments before we reached this decision, and the Reviewing editor has assembled the following comments to help you prepare a revised submission:

1) “We do not know the cause of this repair defect, but one possibility is that allelic DSBs accumulate that have no unbroken sister-chromatid template from which to be repaired. Whatever the cause, cells that form DSBs on partially replicated chromosomes are unable to complete DSB repair.” All sorts of defects could be occurring in these cells, e.g., defective cohesin function (known to cause HR defects). Maybe DSBs continue to be made – indeed, DSB levels in Figure 1 are 40% higher in *cdc6-mn* cells. So, additional deregulation is occurring. Moreover, unlike most mutants with DSB repair defects, checkpoint arrest is not robust in these cells. In addition, with respect to viability loss, surely partially replicated chromosomes are a much bigger problem than unrepaired DSBs? The differences in the amount of replication between *cdc6-mn* and *cdc6-mn spo11* mutants make it impossible to tease out the relative contributions of DSBs and partial replication to cell death. Therefore, it seems that the specific contribution of DSBs made on partially replicated chromosomes to cell death in *cdc6-mn* cells is completely unclear.

2) Figure 6; interhomolog double Holliday structures are detected in *cdc6-mn* cells, so strand exchange can clearly occur. Also, crossover products form at appreciable levels – please show quantification. Does this mean that non-crossover recombination is more defective? Maybe a sister chromatid is important for non-crossover formation? We also wondered whether Holliday structures between sister chromatids are detected in *cdc6-mn*. This is a good control to confirm the replication block in this mutant.

3) There is a lack of quantification throughout the paper, both for Southern and Western analysis. None of the figures have error bars and there's no indication that any of the experiments were done more than once. The authors need to explain how they ensure the reproducibility of their observations.

4) Figure 2; the authors claim that Dbf4 is not hyperphosphorylated in *mec1* strains but this is not clear from the gel that is shown. There is definitely a shift compared to no HU or even *rad53* with HU.

5) Figure 5: the authors claim that the decreased levels of Spo11 observed in the presence of HU are restored in *mec1* and *cdc6-mn* mutants but this is not apparent in the gel that is shown. Quantitation and some loading controls would clarify this issue.

6) The relevance of reduced Spo11 levels with respect to inhibition of DSBs is unclear and claims otherwise should be toned down. DSBs form in wild type cells when Spo11 levels are relatively low so it's not clear that effects detected here are relevant. The authors should look at Spo11 transcript levels (given that *S. pombe* Rec12 transcription was not altered in *S. pombe*).

7) We are also quite suspicious of the rad53-independent effect of *mec1* for the following reason. Rad53 mutants are sicker than *mec1* mutants. This is likely because Rad53 still has a bit of activity from Tel1 activation. If the authors are seeing a non-specific loss of DSBs due to the effect of a loss of all Rad53 activity (in the 53 delete or the *mec1tel1* double), then it is impossible to tell whether the rescue that they see in the *mec1* delete is rad53-dependent. Some Rad53 functions are preserved in a *mec1Δ* and others are not.

8) The data in Figure 3 did not initially strike us as overwhelming. The effect of HU is strong and obvious, but the rescue by *mec1Δ* is very modest. However, it also seems that all of the HU treated lanes are underloaded. The authors never quantify this, but it seems that if you compared to the upper band (or better still integrated over the lane), the rescue by *mec1* would seem better. In fact, the data in Figure 3—figure supplement 1 look better. We think that quantifying this might be helpful. (The quant in Figure 4 is apparently not for HU treatment, although the legend isn't clear. Moreover, we can't see where it says how this was done.)

9) There are a lot of molecular details in the summary figure that are never tested in the manuscript. In fact, the only data that are clear are the ones that are previously known (Dbf4 is a Rad53 substrate and Mer2 is a Cdc7/Dbf4 substrate). The authors should examine the phosphorylation of Rec104 in *cdc7ts*, *mec1Δ*, *rad53Δ*, *mek1Δ*, and the non-phosphorylated *dbf4* alleles.

10) In the section “Mec1 signals to inhibit Mer2 phosphorylation by DDK” subsection of Results:

A) The authors cite Figure 2 regarding the *dbf4* heterozygous allele. However there are no data showing this.

B) Citation of Figure 2—figure supplement 1 regarding *dbf4-Δ71-221* bypassing the replication checkpoint shows Southern data. This needs to be FACS data, as we’re not sure how the data presented addresses this.

11) The phospho-S30 Mer2 data in Figure 2—figure supplement 1 mirrors the total, and therefore it isn't clear if this is just nonspecific. Can the authors use a *cdk* mutant to control this? A phosphosite mutant is not as good, since nonspecific phosphoantibodies still often recognize the exact phosphosite in the unphosphorylated state (but not a mutant site).

12) Unless the authors can override, with specific manipulations (specific mutations etc), the various inhibitory branches of the checkpoint, they cannot formally state that any of them are important to inhibit DSB formation. Therefore, they should be more cautious in their claims about the targets of the checkpoint.

13) Figure 6—figure supplement 1; the data are not very convincing. The SC is presumably abnormal in *cdc6-mn* strain, so how is full SC measured?

14) What is the role of late origin firing for the increased DSB levels seen in WT + HU versus *mec1* + HU scenarios? It would be useful to see a FACS profile for *mec1* + HU.

---

## [Author Response]

*1) “We do not know the cause of this repair defect, but one possibility is that allelic DSBs accumulate that have no unbroken sister-chromatid template from which to be repaired. Whatever the cause, cells that form DSBs on partially replicated chromosomes are unable to complete DSB repair.” All sorts of defects could be occurring in these cells, e.g., defective cohesin function (known to cause HR defects). Maybe DSBs continue to be made – indeed, DSB levels in*
Figure 1
*are 40% higher in* cdc6-mn *cells. So, additional deregulation is occurring. Moreover, unlike most mutants with DSB repair defects, checkpoint arrest is not robust in these cells. In addition, with respect to viability loss, surely partially replicated chromosomes are a much bigger problem than unrepaired DSBs? The differences in the amount of replication between* cdc6-mn *and* cdc6-mn spo11 *mutants make it impossible to tease out the relative contributions of DSBs and partial replication to cell death. Therefore, it seems that the specific contribution of DSBs made on partially replicated chromosomes to cell death in* cdc6-mn *cells is completely unclear*.

The reviewers raise an important point: the viability loss of *cdc6-mn* mutants is likely a compound effect of the fact that these mutants have a severe underlying defect in origin firing and the fact that they form DSBs on unreplicated chromosomes. To address this point, we now included viability analysis of *dbf4-ΔN221/DBF4* cells. The *dbf4-ΔN221* mutation confers a dominant checkpoint bypass (Figure 2). Thus, use of a heterozygous *dbf4-ΔN221/DBF4* strain allowed us to separate the checkpoint function of DBF4from its replication function (as well as other meiotic functions, which appear unaffected in the *dbf4-ΔN221/DBF4* strain). As shown in the revised Figure 6, *dbf4-ΔN221/DBF4* cells exhibit a loss of viability specifically in HU. This loss of viability was entirely dependent on DSB formation, indicating that these mutants form DSBs in HU that are not properly repaired. We therefore concluded that inappropriate DSB formation on replicating chromosomes contributes to the loss of cell viability in checkpoint mutants.

We certainly agree with the reviewers’ point that “all sorts of defects could be occurring” in the *cdc6-mn* mutants with regards to DSB repair and we aimed to convey this in our original statement. The reason we highlighted allelic breaks is that these would be an obvious problem that would uniquely affect unreplicated chromosomes and could explain why some DSBs (at the same locus) were efficiently repaired and others were not. However, we understand that there are many other possible reasons for the repair defect in *cdc6-mn* cells. Thus, we have removed the discussion of speculatory models from this section.

To address the specific possibilities suggested by the reviewers: we do not favor the hypothesis of defective cohesin function, as we have shown previously that *cdc6-mn* cells load Rec8, build axes and form DSBs comparably to wild type cells. By contrast, loss of cohesin causes a very specific loss of axis structure and decreased DSB formation. We also do not have evidence that *cdc6-mn* cells continue to form more DSBs than wild type cells, at least under the conditions that we assayed in this study (*rad50S* mutants), as we do not see the levels of DSBs continually increasing during the time course shown in Figure 1.

*2)*
Figure 6*; interhomolog double Holliday structures are detected in* cdc6-mn *cells, so strand exchange can clearly occur. Also, crossover products form at appreciable levels – please show quantification. Does this mean that non-crossover recombination is more defective? Maybe a sister chromatid is important for non-crossover formation? We also wondered whether Holliday structures between sister chromatids are detected in* cdc6-mn*. This is a good control to confirm the replication block in this mutant*.

We agree that *cdc6-mn* mutants are capable of completing homologous recombination, as evidenced by both recombination intermediates and full recombinant products that can be seen in this one-dimensional gel analysis. However, further analysis of repair pathways is far from straightforward because of the significant and asynchronous DNA replication in this strain (see Figure 6—figure supplement 1), which substantially complicates the analysis and interpretation of 2D gels. First, as is already obvious from the one-dimensional analysis shown in Figure 6, there are likely several different cell populations; some that repair with essentially wild type kinetics and others that fail to make dHJs. This heterogeneity makes more detailed conclusions difficult without extensive further analysis or additional *cdc6* alleles. Second, the asynchronous replication intermediates also obscure major recombination intermediates (most notably single-end invasions), which essentially renders the analysis of these intermediates impossible in standard 2D-gel analysis. As this manuscript is primarily a characterization of the meiotic replication checkpoint, we feel that the in-depth analysis that would be necessary to further dissect DSB repair on partially replicated chromosomes is beyond the scope of this study. Therefore, in this manuscript, we state that there are problems in repair but refrain from making further claims about the types of repair that might be affected.

*3) There is a lack of quantification throughout the paper, both for Southern and Western analysis. None of the figures have error bars and there's no indication that any of the experiments were done more than once. The authors need to explain how they ensure the reproducibility of their observations*.

Although we acknowledge the concern raised by this reviewer, we would like to stress (and now included a note) that all of the Southern and Western blots in this study were performed at least twice on independent experiments with highly reproducible results. Some of the experiments, e.g., DSB levels in *mec1* and *rad53* mutants, were carried out more than 10 times using independent strains and experiments, a subset of which are shown in this paper.

For each Southern blot, we measured relative DSB levels (normalized to the level of DSBs in the wild type strain from the same experiment) for each time point in the time course, as shown in Figures 1 and 3. The quantifications are of the exact blots used in the paper.

To address the lack of quantification of Western data, we now also included quantitative analysis of the Spo11 protein levels. All Spo11 Westerns include titrations of protein examined, to provide a more reliable estimate of relative protein abundance.

*4)*
Figure 2*; the authors claim that Dbf4 is not hyperphosphorylated in* mec1 *strains but this is not clear from the gel that is shown. There is definitely a shift compared to no HU or even* rad53 *with HU*.

We realize that our use of the term “hyperphosphorylation” was confusing in this context. We intended to refer to the extent of phospho-Dbf4 accumulation, e.g., the relative amounts of the protein that run in each band. Dbf4 migrates as multiple bands in all strains, representing varying levels of phosphorylation. When HU is added to wild type cells, more of the Dbf4 protein accumulates in the upper, hyperphosphorylated form than is observed in untreated cells. Although the band representing the hyperphosphorylated protein is still observed in *mec1* and *rad53* mutant cells, much less of the protein accumulates in this form. Therefore, the accumulation of such a large abundance of the total Dbf4 protein in this form is dependent on the checkpoint. We have reworded this section to clarify this point.

*5)*
Figure 5*: the authors claim that the decreased levels of Spo11 observed in the presence of HU are restored in in* mec1 *and* cdc6-mn *mutants but this is not apparent in the gel that is shown. Quantitation and some loading controls would clarify this issue*.

We have now included quantitative Western and Northern analysis of Spo11 protein and RNA levels (new Figure 5 and Figure 5—figure supplement 1). We find that in HU-treated cells, *SPO11* Transcript accumulation abruptly stops at the time of replication onset, ultimately resulting in a roughly 5-fold reduction of *SPO11* RNA. From Western titrations, we estimate that this translates to a >10-fold decrease of Spo11 protein. Furthermore, the new quantifications confirm that this effect is indeed dependent on both Mec1 and Cdc6.

*6) The relevance of reduced Spo11 levels with respect to inhibition of DSBs is unclear and claims otherwise should be toned down. DSBs form in wild type cells when Spo11 levels are relatively low so it's not clear that effects detected here are relevant. The authors should look at Spo11 transcript levels (given that* S. pombe *Rec12 transcription was not altered in* S. pombe*)*.

As indicated in the answer to point 5, we have carried out Northern analysis of *SPO11* RNA levels with and without HU to complement our protein analysis. We found that *SPO11* transcript levels are also decreased, indicating that this regulation occurs at the level of transcription. Therefore, the regulation of Spo11 in budding yeast is distinct from that of Rec12 in fission yeast.

We included quantification indicating that Spo11 protein levels are reduced at least 10-fold upon HU treatment (Figure 5), which likely reduces overall DSB activity. We also show genetically that reducing DSB level scan, in turn, significantly alleviate the replication block caused by DSBs; *cdc6-mn* cells containing hypomorphic alleles of *SPO11* exhibited substantially rescued DNA replication as observed by FACS (Figure 6) and CGH analysis (data not shown). Thus, although Spo11 is clearly not the only checkpoint target, its down-regulation is expected to have beneficial results for survival of meiotic cells with unreplicated DNA. However, we have reworded sections of the manuscript to better clarify that this is a working hypothesis.

*7) We are also quite suspicious of the rad53-independent effect of* mec1 *for the following reason. Rad53 mutants are sicker than mec1 mutants. This is likely because Rad53 still has a bit of activity from Tel1 activation. If the authors are seeing a non-specific loss of DSBs due to the effect of a loss of all Rad53 activity (in the 53 delete or the* mec1tel1 *double), then it is impossible to tell whether the rescue that they see in the* mec1 *delete is rad53-dependent. Some Rad53 functions are preserved in a* mec1Δ *and others are not*.

Because Rad53 can be activated by Tel1, we also analyzed *mec1Δ tel1Δ* mutants in this study (Figure 4—figure supplement 1), which looked similar to *mec1Δ rad53Δ* double mutants. Moreover, the epistasis analysis using the *mec1Δ rad53Δ* cells revealed that the decreased DSB levels in *rad53Δ* cells was not Mec1-dependent and the checkpoint bypass of *mec1Δ* cells was not Rad53-dependent. Therefore, we believe that these are two distinct phenomena, and that they belong in separate genetic pathways.

*8) The data in*
Figure 3
*did not initially strike us as overwhelming. The effect of HU is strong and obvious, but the rescue by* mec1Δ *is very modest. However, it also seems that all of the HU treated lanes are under loaded. The authors never quantify this, but it seems that if you compared to the upper band (or better still integrated over the lane), the rescue by* mec1 *would seem better. In fact, the data in*
Figure 3—figure supplement 1
*look better. We think that quantifying this might be helpful. (The quant in*
Figure 4
*is apparently not for HU treatment, although the legend isn't clear. Moreover, we can't see where it says how this was done.*)

We have now included quantification of the Southern blots in this figure to clarify the extent of DSB formation in *mec1Δ* cells. We also substituted a gel in which the +HU lanes are more equally loaded to compensate for the ∼2-fold reduction in DNA due to the lack of DNA replication in cells treated with HU.

Figure 4 was intended to refer to the decrease in DSB formation that occurs in certain checkpoint mutants in the absence of checkpoint activation. Therefore, we showed only the quantitation for the untreated cells. We also quantified the HU-treated cells in each experiment, but including that data in the graph made it confusing because it was not relevant to the results being discussed in that section. Although we stated in the Results section text that we measured untreated cells, we have also added this to the figure legend to clarify this point.

The quantifications in Figure 4 are of the blots shown in this paper (Figures 2 and 3, Figure 2—figure supplement 1 and Figure 3—figure supplement 1, and Figure 4—figure supplement 1), and the method is now better described in the figure legend.

*9) There are a lot of molecular details in the summary figure that are never tested in the manuscript. In fact, the only data that are clear are the ones that are previously known (Dbf4 is a Rad53 substrate and Mer2 is a Cdc7/Dbf4 substrate). The authors should examine the* phosphorylation *of Rec104 in* cdc7ts*,* mec1Δ*,* rad53Δ*,* mek1Δ*, and the* non-phosphorylated dbf4 *alleles*.

We have shown that in HU-treated cells, Mec1 is responsible for reducing Spo11 transcript and protein levels, Mer2 phosphorylation and Rec114 and Mre11 DNA loading. All of these are novel observations from this study that we believe belong in the model figure.

We marked the potential phosphorylation of Rec104 by Cdc7 with a ‘?’ to indicate that it was a hypothesis. We have removed this arrow completely, as this suggestion is not important to this study and we did not intend to imply that we have shown that Rec104 is a direct target of DDK, which would require mapping and mutating phosphorylation sites on the protein.

*10) In the section “Mec1 signals to inhibit Mer2 phosphorylation by DDK” subsection of Results*:

*A) The authors cite*
Figure 2
*regarding the* dbf4 *heterozygous allele. However there are no data showing this*.

We apologize for this confusion. The wrong version of this figure was uploaded. It has now been replaced by the proper version containing the data referred to in the text.

*B) Citation* of Figure 2—figure supplement 1
*regarding* dbf4-Δ71-221 *bypassing the replication checkpoint shows Southern data. This needs to be FACS data, as we’re not sure how the data presented addresses this*.

In this paper we investigate the role of the meiotic replication checkpoint in preventing DSBs, which is what we refer to as a “checkpoint bypass.” For this reason, we show Southern data in what is now Figure 2—figure supplement 1. We would like to emphasize that we used FACS analysis for every experiment to ensure that any significant DSB formation in HU-treated cells occurred in the absence of full DNA replication, thus separating the stalling of replication forks from the formation of DSBs on unreplicated chromosomes (as is seen for the mec1 checkpoint-deficient strains in Figure 3 and Figure 3—figure supplement 1).

*11) The phospho-S30 Mer2 data in*
Figure 2—figure supplement 1
*mirrors the total, and therefore it isn't clear if this is just nonspecific. Can the authors use a* cdk *mutant to control this? A phosphosite mutant is not as good, since nonspecific phosphoantibodies still often recognize the exact phosphosite in the unphosphorylated state (but not a mutant site)*.

The control experiment comparing wild type and *clb5Δ clb6Δ* cells lacking prophase CDK activity was carried out and is included in the new Figure 2—figure supplement 1. This antibody was also used in [40] with similar specificity.

*12) Unless the authors can override, with specific manipulations (specific mutations etc), the various inhibitory branches of the checkpoint, they cannot formally state that any of them are important to inhibit DSB formation. Therefore, they should be more cautious in their claims about the targets of the checkpoint*.

We agree that we cannot say to what extent the checkpoint-dependent regulation of DSB factors directly inhibits DSB formation. However, we know from the published literature that phosphorylation of Mer2 by DDK is essential for DSB formation, so we can predict that failure to phosphorylate Mer2 would have a significant impact on DSB formation. Similarly, we believe that down-regulating Spo11 significantly improves the DNA replication block experienced by *cdc6-mn* cells (see point 6 above). It is not currently clear in the field whether Mre11 and Rec114 must be on chromosomes to allow for DSB formation, however this is a likely prediction and prevailing model in the field (see [38]). Therefore, it seems reasonable to hypothesize that all three methods of DSB factor regulation described in this study contribute significantly to preventing lethal DSB formation on partially replicated chromosomes. However, we have reworded the Abstract, Results, and Discussion sections of the paper to reflect the point that we cannot formally state that the observed regulation of DSB factors inhibits DSB formation.

*13)*
Figure 6—figure supplement 1*; the data are not very convincing. The SC is presumably abnormal in* cdc6-mn *strain, so how is full SC measured*?

In fact, SC formation in *cdc6-mn* cells occurs relatively efficiently, and previous analysis actually failed to detect defective SC assembly in these cells (6). However, closer examination revealed that *cdc6-mn* cells exhibit an increase in the number of cells with regions of the nucleus without full SC assembly, which is what we counted. Image samples of a wild type cell with full SC and a cdc6-mn cell with partial SC and a polycomplex are shown in Figure 6—figure supplement 1 and the graph in Figure 6—figure supplement 1 is the result of counting 200 cells for each strain and time point. We have attempted to further clarify this point and we cite the work of [6] at this point in the manuscript.

*14) What is the role of late origin firing for the increased DSB levels seen in WT + HU versus* mec1 *+ HU scenarios? It would be useful to see a FACS profile for* mec1 *+ HU*.

We have now included the FACS profile for all of the strains in Figure 3, in addition to the FACS for *mec1Δ* cells that was already in Figure 3—figure supplement 1. These data indicate that there is no significant difference in genomic replication between *mec1* and *MEC1* cells. As mentioned above, we performed FACS analysis on all of the strains studied in this paper to ensure that DSBs in HU formed in the absence of increased DNA replication.

To further address the question of whether increased DNA replication contributes to the DSBs formed in *mec1Δ* cells +HU, we also probed a late replicating chromosomal region that would not be expected to initiate replication even in *mec1Δ* cells, and saw the same checkpoint bypass as at the *yCR047* locus shown in Figure 3. Thus, there is currently no evidence that the DSBs we observed in *mec1Δ* cells are due to increased DNA replication in these cells. This experiment is currently not included in the revised manuscript as it appears redundant with our PFGE analysis of *mec1Δ* cells +HU, which revealed that DSBs formed across the genome without significant DNA replication (Figure 3—figure supplement 1).